# Enhancing the Maximum Effective Window for Long-Term Time Series Forecasting

**Jiahui Zhang**[1,2], **Zhengyang Zhou**[1,2], **Wenjie Du**[1,2,†], **Yang Wang**[1,2,†]

[1]University of Science and Technology of China, China
[2]Suzhou Institute for Advanced Research, USTC, China

kongping@mail.ustc.edu.cn   {zzy0929, duwenjie, angyan}@ustc.edu.cn

## Abstract

Long-term time series forecasting (LTSF) aims to predict future trends based on historical data. While longer lookback windows theoretically offer more comprehensive insights, Transformer-based models often struggle with them. On one hand, longer windows introduce more noise and redundancy, hindering the model's learning process. On the other hand, Transformers suffer from attention dispersion and are prone to overfitting to noise, especially when processing long sequences. In this paper, we introduce the Maximum Effective Window (MEW) metric to assess a model's ability to effectively utilize the lookback window. We also propose two model-agnostic modules to enhance MEW, enabling models to better leverage historical data for improved performance. Specifically, to reduce redundancy and noise, we introduce the Information Bottleneck Filter (IBF), which employs information bottleneck theory to extract the most essential subsequences from the input. Additionally, we propose the Hybrid-Transformer-Mamba (HTM), which incorporates the Mamba mechanism for selective forgetting of long sequences while harnessing the Transformer's strong modeling capabilities for shorter sequences. We integrate these two modules into various Transformer-based models, and experimental results show that they effectively enhance MEW, leading to improved overall performance. Our code is available at https://github.com/forever-ly/PIH.

## 1 Introduction

Long-term time series forecasting (LTSF) [14] holds significant importance across various domains such as traffic management, energy optimization, and financial analysis. Transformer-base methods [31], known for their attention mechanisms that facilitate the automatic learning of sequential dependencies, have emerged as promising tools for LTSF.

Intuitively, extending the lookback window is a natural choice to enhance the forecasting capability of the model. This allows the model to capture long-term trends more accurately, thereby improving its ability to predict seasonal variations, cyclical patterns, and overall trends. However, prior research has found that current Transformer-based models are not effective in leveraging long lookback windows [40]. To quantitatively assess a model's efficiency in utilizing the lookback window, we propose a new metric **Maximum Effective Window (MEW):** *For a given model, as the lookback window is increased while keeping other settings constant, there exists a point beyond which further increases in the window do not result in better performance. This point is referred to as the model's Maximum Effective Window, which reflects the model's potential to utilize historical information.*

---

† : corresponding author

39th Conference on Neural Information Processing Systems (NeurIPS 2025).

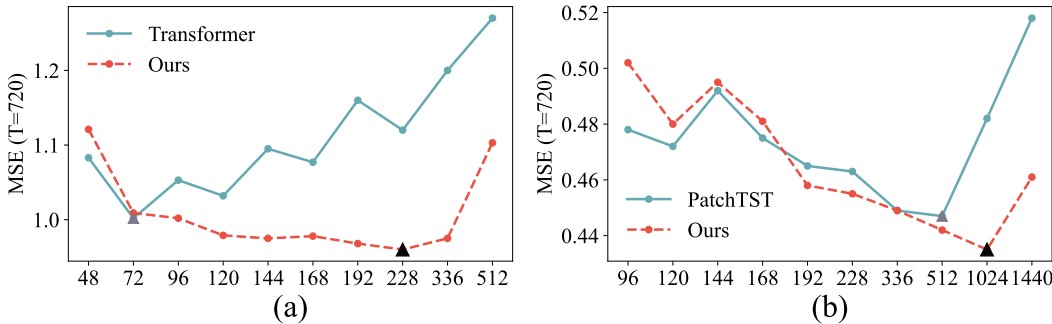

Figure 1: The changes in MSE with respect to the lookback window for Transformer and PatchTST on the *ETTh1* dataset with a prediction length of $T = 720$. "Ours" refers to the integration of our modules into these two models. The triangular markers indicate the Maximum Effective Window (MEW). "Ours" increases the MEW from 72 and 512 to 288 and 1024, respectively.

As shown in Fig. 1 (a), point-based Transformer (where a single time point is treated as a token) only achieve MEW of 72 for the *ETTh1* dataset. A natural question then arises: *Can we increase a model's MEW, thereby enabling it to perform better with longer lookback windows?* Before answering this, we first analyze the reasons behind the limitation of the MEW. **From the information perspective**, longer windows may contain more redundant signals [26], which can impede the model's learning process by allocating significant modeling capacity to meaningless or irrelevant noise signals [40]. **From the architecture perspective**, many studies have shown that Transformers suffer from issues such as attention dispersion and overfitting to noise [37], and these drawbacks are exacerbated with longer sequences.

The patch strategy [23, 41] is an effective approach to improving the MEW. By treating consecutive time points as a single patch, this strategy significantly reduces local-level noise whihin a patch. Additionally, it greatly reduces the number of tokens that the Transformer process, it helps mitigate issues such as attention dispersion and overfitting to noise. As shown in Fig. 1 (b), PatchTST achieves a MEW of 512, which is substantially higher than the 72 achieved by point-based Transformer. However, the patch strategy is heuristic and lacks adaptability. Firstly, it assumes redundancy and noise only within the points of each patch, which is a heuristic assumption that overlooks potential redundancy and noise between patches. Secondly, when the number of patches increases, the issues of attention dispersion and overfitting to noise in Transformers can still be exacerbated. Lastly, this approach is not suited for models that do not adopt a patch-based design.

In this paper, we propose two model-agnostic modules to address the issues of information redundancy and architectural limitations. As shown in Fig. 1, after integrating our modules, the MEW for PatchTST (patch-based) and Vanilla Transformer (point-based) increases from 72 and 512 to 288 and 1024, respectively, achieving improved performance.

Specifically, **to reduce redundancy and noise in long lookback windows, we introduce the Information Bottleneck Filter (IBF) module based on Information Bottleneck (IB) theory**. This module aims to identify informative subsequences while minimizing redundancy and noise [3], enabling the model to prioritize significant subsequences within the lookback windows. However, directly optimizing the IB objective for sequences is challenging due to their discrete nature [39, 38], often leading to training instability and degraded predictions. To address this, we propose a probabilistic framework for sequence selection, coupled with a noise injection strategy. The core idea is that important subsequences should have a low probability of noise injection, whereas injecting larger noise into redundant sequences has minimal impact on predictions. By tailoring a noise prior for each input, the IB objective can yield a manageable variational upper bound. To overcome the challenges of attention dispersion and overfitting to noise in Transformers, we introduce Mamba [7], a state-space model (SSM) capable of selectively remembering historical information and filtering out noise. Recent studies have suggested that Transformers and SSMs complement each other in modeling [21, 24]. Therefore, rather than replacing Transformers with Mamba, we propose a Hybrid-Transformer-Mamba (HTM) architecture. First, Mamba captures long-term information while selectively filtering out noise from the extended lookback window. Then, the IBF further reduces noise. Finally, leveraging the temporal characteristics of the time series, we introduce two sequence

splitting algorithms—interval split and block split—to divide the noise-filtered long lookback window into multiple short subsequences. These subsequences are processed by the Transformer to capture short-term dependencies. Experimental results show that HTM outperforms both pure Transformer and pure Mamba architectures in terms of both computational overhead and predictive performance. In summary, our primary contributions are as follows:

1. We defined a metric MEW, which reflects the model's ability to leverage the window effectively. We analyzed how to improve MEW of Transformer-based models from both the information-theoretic and architectural perspectives, and proposed two model-agnostic modules: IBF and HTM.

2. We integrated these two modules into multiple Transformer-based models and conducted detailed experiments on seven datasets. The results demonstrate that these modules can enhance MEW while achieving better performance.

3. Notably, by incorporating these modules into the PatchTST model, we developed the PIH model (**P**atch-**I**BF-**H**TM), where the window length is extended to 1024—surpassing the window settings of all currently existing non-LLM-based time series models, to the best of our knowledge. The PIH model achieved state-of-the-art results, demonstrating the effectiveness of enhancing the model's windows. Our work paves the way for future research to explore even longer window sizes.

## 2   Related Work

**Transformer-based Models.**   Early attempts [28, 19, 11] at directly applying vanilla Transformers to time series data failed in long sequence forecasting tasks, as the self-attention operation scales quadratically with the input sequence length. Existing approaches primarily address this challenge through two avenues. Patch-based methods, exemplified by PatchTST [23] and CrossFormer [41], conceptualize consecutive time steps as patches, reducing the number of input tokens and augmenting local semantics to mitigate redundancy. Another approach focuses on sparse attention mechanisms. Models such as Informer [42], Autoformer [35], Pyraformer [17], and FEDformer [43] adapt the self-attention mechanism to achieve complexities of $O(L)$ or $O(L \log(L))$. These models rely on specific designs and often sacrifice representational capacity, thereby compromising performance. Our work is independent of these approaches and can be effectively integrated into them.

**Mamba for Time Series.**   Recently, several approaches have emerged to incorporate Mamba into time series modeling. Bi-Mamba+ [12] introduces a novel Mamba+ block by incorporating a forget gate within Mamba. This modification enables the selective combination of new features with historical ones in a complementary manner, boosting the model's ability to balance past and present information. S-Mamba [34] adopts a different approach by autonomously tokenizing time points of each variate using a linear layer. The method employs a bidirectional Mamba layer to extract inter-variate correlations and a Feed-Forward Network to learn temporal dependencies. TimeMachine [2] takes a broader view of time series data by leveraging multi-scale contextual cues. Its architecture integrates a quadruple-Mamba design, allowing the model to manage both channel-mixing and channel-independence scenarios. MambaTS [4] challenges the necessity of causal convolution within Mamba for LTSF. It proposes the Temporal Mamba Block (TMB) as an alternative. To further prevent model overfitting, MambaTS incorporates a dropout mechanism that selectively applies to TMB's parameters, ensuring a more stable and generalizable model performance.

**Information Bottleneck (IB).**   The essence of the IB principle lies in distilling a compact yet predictive code from the input signal [30]. Pioneering work by  [3] introduced the concept of variational information bottleneck (VIB), thereby enriching deep learning methodologies. Given random variables $X$ and $Y$, IB aims to compress $X$ into a bottleneck random variable $B$, while retaining information pertinent to predicting $Y$:

$$\min_{B} -I(Y; B) + \beta I(X; B) \tag{1}$$

Here, $\beta$ serves as a Lagrangian multiplier to balance the two mutual information terms. Presently, IB and VIB find extensive applications in deep learning, predominantly in representation learning and feature selection domains [8, 33, 36]. Strategies such as injecting noise into intermediate

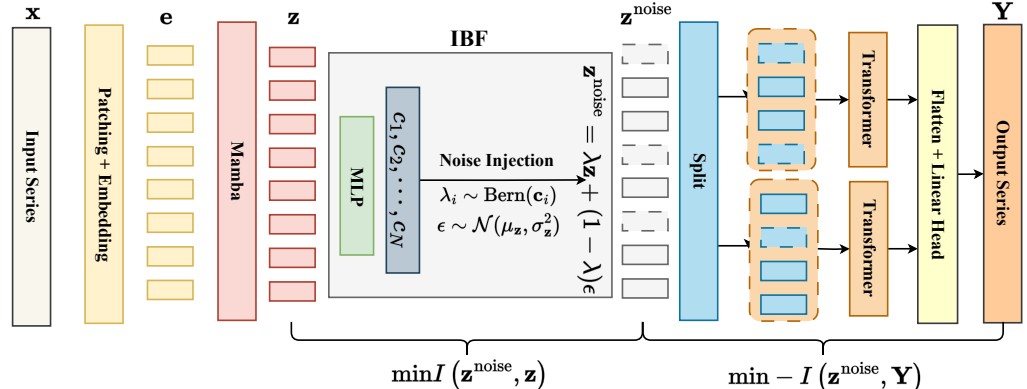

Figure 2: Overview of the PIH. The input long sequence $\mathbf{x}$ first undergoes patching and embedding to obtain $\mathbf{e}$, which is then processed by Mamba to reduce redundancy and noise based on its selective mechanism, resulting in $\mathbf{z}$. The IBF evaluates the importance $\mathbf{c}$ of tokens in the sequence $\mathbf{z}$ and adaptively injects noise to produce $\mathbf{z}^{\text{noise}} = \lambda \mathbf{z} + (1 - \lambda)\epsilon$, where $\lambda$ is sampled from a Bernoulli distribution with parameter $\mathbf{c}$, and $\epsilon$ is random Gaussian noise. The tokens within the dashed box represent less important tokens, with corresponding $\lambda_i$ values being smaller, thus $\mathbf{z}_i^{\text{noise}}$ is primarily controlled by random noise $(1 - \lambda_i)\epsilon$. Conversely, the important tokens within the solid box correspond to larger $\lambda_j$, with their values being mainly determined by their inherent semantics $\lambda_j \mathbf{z}_j$. After splitting $\mathbf{z}^{\text{noise}}$ into multiple subsequences, they are processed by the Transformer. Finally, the output is passed through flattening and a linear head to obtain the prediction.

representations of pre-trained networks and subsequently selecting regions with optimal information per dimension have been explored [1, 27].

## 3 Method

We integrate HTM and IBF into the PatchTST, resulting in the PIH model (**P**atch-**I**BF-**H**TM), as illustrated in Fig. 2.

**Problem Definition.** Given a collection of multivariate time series samples with a lookback window $L = (\mathbf{x}_1, \ldots, \mathbf{x}_L)$, where each $\mathbf{x}_t$ at time step $t$ is a vector of dimension $C$, we aim to forecast $T$ future values $(\mathbf{x}_{L+1}, \ldots, \mathbf{x}_{L+T})$.

### 3.1 Patching and Embedding

Given our utilization of a channel-independent strategy [23], we opt for simplicity by converting multivariate time series into univariate ones. The input univariate time series $\mathbf{x} \in \mathbb{R}^{1 \times L}$ is initially segmented into patches, which may be either overlapping or non-overlapping. Employing patching strategies enhances locality and captures comprehensive semantic information beyond the point level by aggregating time steps into subseries-level patches. Furthermore, to ensure uniform partitioning of the patch sequence into $K$ equally-sized blocks in subsequent modules (see Section 3.3), we employ $\text{Padding}(\cdot)$ to extend the input sequence. Denoting the patch length as $P$ and the stride (the non-overlapping region between two consecutive patches) as $S$, the $\text{Patch}(\cdot)$ process yields a sequence of patches $\mathbf{h} \in \mathbb{R}^{N \times P}$, where $N$ denotes the number of patches, $N = \lceil \frac{(L-P)}{SK} \rceil * K$. Subsequently, we employ an embedding layer to map the dimension of each patch from $\mathbf{h} \in \mathbb{R}^{N \times P}$ to $\mathbf{e} \in \mathbb{R}^{N \times d}$.

$$\mathbf{e} = \text{Embedding}\left(\text{Patch}\left(\text{Padding}\left(\mathbf{x}\right)\right)\right) \tag{2}$$

After obtaining the patch embedding sequence $\mathbf{e} = \{e_1, e_2, \ldots, e_N\}$, we use Mamba followed by a subsequent Dropout layer to capture long-term dependency:

$$\mathbf{z} = \text{Dropout}(\text{Mamba}(\mathbf{e})) \tag{3}$$

Mamba can selectively pass or forget information, which helps reduce redundancy and noise in the input $\mathbf{e}$. However, relying solely on Mamba is not sufficient. In scenarios where the patch sequence length $N$ is large, there still exists the possibility of significant redundancy. Therefore, we propose the IBF to further filter out redundant information from $\mathbf{z}$.

## 3.2 IBF Module for Redundancy Filtering

The IBF module seeks to retrieve the most relevant subsequence $\mathbf{x}^{\text{sub}}$ for a target prediction $\mathbf{Y}$ from the input sequence $\mathbf{x}$. We adopt the sufficient encoder assumption [29], implying that the information of the input subsequence $\mathbf{x}^{\text{sub}}$ is preserved in the encoding process, resulting in $I(\mathbf{x}^{\text{sub}}, \mathbf{Y}) \approx I(\mathbf{z}^{\text{sub}}, \mathbf{Y})$ and $I(\mathbf{x}^{\text{sub}}, \mathbf{x}) \approx I(\mathbf{z}^{\text{sub}}, \mathbf{z})$, where $\mathbf{z}^{\text{sub}}$ is a subsequence of $\mathbf{z}$. The Eq. 1 are transformed into:

$$\min_{\mathbf{z}^{\text{sub}}} -I(\mathbf{z}^{\text{sub}}, \mathbf{Y}) + \beta I(\mathbf{z}^{\text{sub}}, \mathbf{z}) \tag{4}$$

The first term encourages $\mathbf{z}^{\text{sub}}$ to be informative with respect to the label $\mathbf{Y}$, and the second term minimizes the mutual information between $\mathbf{z}$ and $\mathbf{z}^{\text{sub}}$, so that $\mathbf{z}^{\text{sub}}$ retains only limited information from $\mathbf{z}$. The discrete nature of sequences renders direct optimization of IB objective impractical, as there are $2^N$ potential subsequences $\mathbf{z}^{\text{sub}}$ for a patch sequence of length $N$. Here, we relax patch selection from discrete to probabilistic sampling. Considering $\mathbf{z}_i$ as the representation of the $i$-th patch, we utilize $\text{MLP}(\cdot)$ to assess the importance $\mathbf{c}_i$ of patch $\mathbf{z}_i$:

$$\mathbf{c}_i = \text{sigmoid}(\text{MLP}(\mathbf{z}_i)) \tag{5}$$

Consequently, the selection of patch $\mathbf{z}_i$ can be obtained by sampling from $\lambda_i \sim \text{Bern}(\mathbf{c}_i)$, where $\text{Bern}(\mathbf{c}_i)$ represents a Bernoulli distribution parameterized by $\mathbf{c}_i$. To ensure the differentiability of the sampling process, we utilize the gumbel sigmoid [20, 9] function for the discrete random variable $\lambda_i$, defined as:

$$\lambda_i = \text{Sigmoid}\left(\frac{1}{\tau} \log\left[\frac{\mathbf{c}_i}{1 - \mathbf{c}_i}\right] + \log\left[\frac{u}{1 - u}\right]\right) \tag{6}$$

where $u \sim \text{Uniform}(0, 1)$, and $\tau$ is the temperature. Subsequently, subsequence $\mathbf{z}^{\text{sub}}$ can be obtained by $\mathbf{z}^{\text{sub}} = \lambda \mathbf{z}$.

Although we can employ shannon mutual information [5] to quantify the compressed and informative nature of the distribution of subsequences $\mathbf{z}^{\text{sub}}$, the optimization process is inefficient and unstable due to mutual information estimation [39]. Here, we employ an optimization strategy known as noise injection [38], which endows the IB objective with a tractable variational upper bound. The core concept is to allow the model to introduce noise into less informative subsequences while minimizing noise injection into more informative ones (i.e., $\mathbf{z}^{\text{sub}}$). Initially, noise injection disrupts the flow of information from the input sequence $\mathbf{z}$ to the perturbed sequence $\mathbf{z}^{\text{noise}} = \lambda \mathbf{z} + (1 - \lambda)\epsilon$, where $\epsilon$ follows a random Gaussian distribution. To preserve the semantic, we set $\epsilon \sim \mathcal{N}(\mu_{\mathbf{z}}, \sigma_{\mathbf{z}}^2)$, where $\mu_{\mathbf{z}}$ and $\sigma_{\mathbf{z}}^2$ denote the mean and variance of $\mathbf{z}$. Subsequently, we encourage the perturbed sequence $\mathbf{z}^{\text{noise}}$ to maintain its informative properties relative to the label $\mathbf{Y}$. Finally, $\mathbf{z}^{\text{sub}}$ is derived by removing the noise from $\mathbf{z}^{\text{noise}}$. Eq. 4 can be reformulated as:

$$\min_{\mathbf{z}^{\text{noise}}} -I(\mathbf{z}^{\text{noise}}, Y) + \beta I(\mathbf{z}^{\text{noise}}, \mathbf{z}) \tag{7}$$

**Minimizing** $-I(\mathbf{z}^{\text{noise}}, \mathbf{Y})$**.** We first examine the first term $-I(\mathbf{z}^{\text{noise}}, \mathbf{Y})$ in Eq. 7 which encourages $\mathbf{z}^{\text{noise}}$ is informative of label $\mathbf{Y}$:

$$-I(\mathbf{z}^{\text{noise}}, \mathbf{Y}) \leq \mathbb{E}_{\mathbf{Y}, \mathbf{z}^{\text{noise}}} -\log p_\theta(\mathbf{Y} \mid \mathbf{z}^{\text{noise}}) := \mathcal{L}_{\text{pred}}(\mathbf{z}^{\text{noise}}, \mathbf{Y}) \tag{8}$$

Here, $p_\theta(\mathbf{Y} \mid \mathbf{z}^{\text{noise}})$ represents the variational approximation to the true posterior distribution $p(\mathbf{Y} \mid \mathbf{z}^{\text{noise}})$ (A detailed proof can be found in Appendix D). We model $p_\theta(\mathbf{Y} \mid \mathbf{z}^{\text{noise}})$ as a predictor parametrized by $\theta$, which outputs the model prediction $\mathbf{Y}$ based on the input $\mathbf{z}^{\text{noise}}$. Thus, we can minimize the upper bound of $-I(\mathbf{z}^{\text{noise}}, \mathbf{Y})$ by minimizing the model prediction loss $\mathcal{L}_{\text{pred}}(\mathbf{z}^{\text{noise}}, \mathbf{Y})$. We choose to utilize the MSE as $\mathcal{L}_{\text{pred}}(\mathbf{z}^{\text{noise}}, \mathbf{Y})$.

**Minimizing $I(\mathbf{z}^{\mathbf{noise}}, \mathbf{z})$.** For the second term $I(\mathbf{z}^{\text{noise}}, \mathbf{z})$ in Eq. 7, we can derive its variational upper bound:

$$-I\left(\mathbf{z}^{\text{noise}}, \mathbf{z}\right) \leq \mathbb{E}_{\mathbf{z}}\left(-\frac{1}{2}\log A + \frac{1}{2N}A + \frac{1}{2N}B^2\right) := \mathcal{L}_{\text{comp}}\left(\mathbf{z}^{\text{noise}}, \mathbf{z}\right) \qquad (9)$$

where $A = \sum_{j=1}^{N}\left(1 - \lambda_j\right)^2$ and $B = \frac{\sum_{j=1}^{N}\lambda_j(\mathbf{z}_j - \mu_{\mathbf{z}})}{\sigma_{\mathbf{z}}}$. A detail proof is given in Appendix D.

Finally, we can efficiently estimate Eq. 8 and Eq. 9 with the batched data in the training set. The overall loss is:

$$\mathcal{L} = \mathcal{L}_{\text{pred}}\left(\mathbf{z}^{\text{noise}}, \mathbf{Y}\right) + \beta\mathcal{L}_{\text{comp}}\left(\mathbf{z}^{\text{noise}}, \mathbf{z}\right) \qquad (10)$$

### 3.3 Hybrid-Transformer-Mamba(HTM)

Modeling the input long sequence with Mamba and then using Transformer to model the partitioned short sequences is a promising paradigm [22, 25, 13], as it can leverage the strengths of both architectures simultaneously. We have designed two split methods capable of retaining semantic information: *interval split* and *block split*, denoted as:

$$b_i = \{\mathbf{z}_j^{\text{noise}} \in \mathbf{z}^{\text{noise}} : i \equiv j \pmod{K}\} \qquad (11)$$

$$b_i = \mathbf{z}_{[(i-1)*N/K:i*N/K]}^{\text{noise}} \qquad (12)$$

where $b_i$ represents the $i$-th sequence block, and $K$ is the number of blocks. The premise for splitting sequences into subsequences is that the latter can still retain the semantic meaning of the original long sequences. Fortunately, time series data often adhere to this principle. The *interval split* is inspired by SCINet [15], which highlights a unique property of time series: temporal relations (e.g., trend and seasonal components) are largely preserved after downsampling into two subsequences. SCINet downsamples the original sequence into two subsequences by separating the even and odd elements, our *interval split* extends this approach to partitioning sequence into $K$ blocks, distributing contiguous $K$ patches into $K$ distinct blocks. This method preserves the global characteristics of the sequence. Additionally, we propose the *block split*, where a continuous segment of patch subsequence forms a block. This partitioning method is based on the periodicity of time series, where one period (or multiples of a period) is considered as a block, thus preserving the local information of the sequence. The patch operation and partitioning reduce the length of the input sequence for the Transformer from $L$ to $L/PK$, significantly reducing the computational overhead. For short sequences, the Transformer is highly efficient, even more so than Mamba (see Appendix E.2).

## 4 Experiments

Our experiments aim to address the following questions:

- Can the PIH model effectively reduce the noise in the input sequence and thus leverage a longer lookback window? (Section 4.1)
- Can our approach universally improve the MEW of various Transformer-based models and achieve better performance? (Section 4.2)
- What is the impact of each component? (Section 4.3)

### 4.1 PIH Can Effectively Leverage Long Windows

**Dataset.** We evaluate PIH on seven popular datasets, including Weather, Traffic, Electricity, and four ETT datasets (ETTh1, ETTh2, ETTm1, ETTm2). We also provide results on the Solar and PEMS datasets in Appendix C.1.

**Experimental Settings and Baselines.** PIH integrates the IBF and HTM modules into the PatchTST model, making PatchTST the primary baseline. To assess how effectively our model utilizes longer lookback windows, we set $L = 1024$ for both PIH and PatchTST, which is significantly longer than in previous studies. The other experimental settings can be found in Appendix 2. We additionally

| Models | PIH Ours | | PatchTST 2023 | | Bi-Mamba 2024 | | S-Mamba 2024 | | FEDformer 2022 | | Autoformer 2021 | | Informer 2020 | | DLinear 2023 | | NLinear 2023 | |
|---|---|---|---|---|---|---|---|---|---|---|---|---|---|---|---|---|---|---|
| Metric | MSE | MAE | MSE | MAE | MSE | MAE | MSE | MAE | MSE | MAE | MSE | MAE | MSE | MAE | MSE | MAE | MSE | MAE |
| ETTh1 96 | **0.360** | **0.394** | 0.371 | 0.405 | 0.378 | 0.395 | 0.386 | 0.406 | 0.376 | 0.415 | 0.435 | 0.446 | 0.941 | 0.769 | 0.511 | 0.520 | 0.379 | 0.404 |
| ETTh1 192 | **0.396** | **0.418** | 0.408 | 0.429 | 0.427 | 0.428 | 0.448 | 0.444 | 0.423 | 0.446 | 0.456 | 0.457 | 1.007 | 0.786 | 0.414 | 0.428 | 0.414 | 0.426 |
| ETTh1 336 | **0.409** | **0.432** | 0.431 | 0.449 | 0.471 | 0.445 | 0.494 | 0.468 | 0.444 | 0.462 | 0.486 | 0.487 | 1.038 | 0.784 | 0.453 | 0.458 | 0.442 | 0.445 |
| ETTh1 720 | **0.435** | **0.466** | 0.482 | 0.483 | 0.470 | 0.457 | 0.493 | 0.488 | 0.469 | 0.492 | 0.515 | 0.517 | 1.144 | 0.857 | 0.511 | 0.520 | 0.470 | 0.477 |
| ETTh2 96 | **0.263** | **0.328** | 0.277 | 0.340 | 0.291 | 0.342 | 0.298 | 0.349 | 0.332 | 0.374 | 0.332 | 0.368 | 1.549 | 0.952 | 0.294 | 0.361 | 0.296 | 0.351 |
| ETTh2 192 | **0.324** | **0.370** | 0.343 | 0.385 | 0.368 | 0.392 | 0.379 | 0.398 | 0.407 | 0.446 | 0.426 | 0.434 | 3.792 | 1.542 | 0.430 | 0.448 | 0.337 | 0.382 |
| ETTh2 336 | **0.314** | **0.376** | 0.338 | 0.394 | 0.407 | 0.424 | 0.417 | 0.432 | 0.400 | 0.447 | 0.477 | 0.479 | 4.215 | 1.642 | 0.492 | 0.484 | 0.359 | 0.407 |
| ETTh2 720 | **0.378** | **0.425** | 0.403 | 0.442 | 0.421 | 0.439 | 0.431 | 0.449 | 0.412 | 0.469 | 0.453 | 0.490 | 3.656 | 1.619 | 0.905 | 0.683 | 0.417 | 0.456 |
| ETTm1 96 | **0.291** | **0.349** | 0.294 | 0.349 | 0.320 | 0.360 | 0.331 | 0.368 | 0.326 | 0.390 | 0.510 | 0.492 | 0.626 | 0.560 | 0.314 | 0.358 | 0.317 | 0.359 |
| ETTm1 192 | 0.337 | **0.374** | **0.334** | **0.374** | 0.361 | 0.383 | 0.371 | 0.387 | 0.365 | 0.415 | 0.514 | 0.495 | 0.725 | 0.619 | 0.356 | 0.391 | 0.352 | 0.381 |
| ETTm1 336 | **0.360** | **0.386** | 0.363 | 0.392 | 0.386 | 0.402 | 0.417 | 0.418 | 0.392 | 0.425 | 0.510 | 0.492 | 1.005 | 0.741 | 0.365 | 0.388 | 0.374 | 0.393 |
| ETTm1 720 | **0.405** | **0.411** | 0.407 | 0.416 | 0.445 | 0.437 | 0.471 | 0.448 | 0.446 | 0.458 | 0.527 | 0.493 | 1.133 | 0.845 | 0.410 | 0.417 | 0.409 | 0.413 |
| ETTm2 96 | **0.161** | **0.253** | 0.164 | 0.259 | 0.176 | 0.263 | 0.179 | 0.263 | 0.180 | 0.271 | 0.205 | 0.293 | 0.355 | 0.462 | 0.164 | 0.260 | 0.163 | 0.257 |
| ETTm2 192 | **0.213** | **0.289** | 0.216 | 0.295 | 0.242 | 0.304 | 0.253 | 0.310 | 0.252 | 0.318 | 0.278 | 0.336 | 0.595 | 0.586 | 0.238 | 0.317 | 0.216 | 0.294 |
| ETTm2 336 | **0.265** | **0.326** | 0.268 | 0.331 | 0.304 | 0.344 | 0.312 | 0.348 | 0.324 | 0.364 | 0.343 | 0.379 | 1.270 | 0.871 | 0.265 | 0.326 | 0.265 | 0.326 |
| ETTm2 720 | **0.342** | **0.375** | 0.350 | 0.383 | 0.402 | 0.402 | 0.412 | 0.408 | 0.410 | 0.420 | 0.414 | 0.419 | 3.001 | 1.267 | 0.338 | 0.375 | 0.338 | 0.375 |
| Weather 96 | **0.147** | 0.198 | **0.147** | **0.197** | 0.159 | 0.205 | 0.166 | 0.210 | 0.238 | 0.314 | 0.249 | 0.329 | 0.354 | 0.405 | 0.167 | 0.225 | 0.170 | 0.226 |
| Weather 192 | 0.191 | **0.239** | **0.190** | 0.241 | 0.205 | 0.249 | 0.215 | 0.253 | 0.275 | 0.329 | 0.325 | 0.370 | 0.419 | 0.434 | 0.211 | 0.267 | 0.215 | 0.265 |
| Weather 336 | **0.241** | **0.280** | 0.243 | 0.283 | 0.264 | 0.291 | 0.276 | 0.298 | 0.339 | 0.377 | 0.351 | 0.391 | 0.583 | 0.543 | 0.255 | 0.304 | 0.259 | 0.298 |
| Weather 720 | 0.309 | 0.329 | **0.306** | **0.328** | 0.343 | 0.344 | 0.353 | 0.349 | 0.389 | 0.409 | 0.415 | 0.426 | 0.916 | 0.705 | 0.313 | 0.351 | 0.321 | 0.342 |
| Traffic 96 | **0.357** | **0.248** | 0.394 | 0.289 | 0.375 | 0.258 | 0.381 | 0.261 | 0.576 | 0.359 | 0.597 | 0.371 | 0.733 | 0.410 | 0.385 | 0.275 | 0.383 | 0.270 |
| Traffic 192 | **0.371** | **0.255** | 0.407 | 0.295 | 0.394 | 0.269 | 0.397 | 0.267 | 0.610 | 0.380 | 0.607 | 0.382 | 0.777 | 0.435 | 0.397 | 0.279 | 0.397 | 0.274 |
| Traffic 336 | **0.392** | **0.261** | 0.422 | 0.302 | 0.406 | 0.274 | 0.423 | 0.276 | 0.608 | 0.375 | 0.623 | 0.387 | 0.776 | 0.434 | 0.412 | 0.288 | 0.410 | 0.281 |
| Traffic 720 | **0.430** | **0.282** | 0.46 | 0.319 | 0.440 | 0.288 | 0.458 | 0.300 | 0.621 | 0.375 | 0.639 | 0.395 | 0.827 | 0.466 | 0.450 | 0.309 | 0.449 | 0.303 |
| Electricity 96 | **0.127** | **0.220** | 0.133 | 0.226 | 0.140 | 0.238 | 0.142 | 0.238 | 0.186 | 0.302 | 0.196 | 0.313 | 0.304 | 0.393 | 0.132 | 0.229 | 0.133 | 0.229 |
| Electricity 192 | **0.145** | **0.240** | 0.151 | 0.249 | 0.155 | 0.253 | 0.169 | 0.267 | 0.197 | 0.311 | 0.211 | 0.324 | 0.327 | 0.417 | 0.146 | 0.243 | 0.148 | 0.242 |
| Electricity 336 | **0.160** | **0.256** | 0.167 | 0.263 | 0.170 | 0.269 | 0.178 | 0.275 | 0.213 | 0.328 | 0.214 | 0.327 | 0.333 | 0.422 | 0.161 | 0.260 | 0.164 | 0.259 |
| Electricity 720 | **0.192** | **0.287** | 0.206 | 0.299 | 0.196 | 0.293 | 0.207 | 0.303 | 0.233 | 0.344 | 0.236 | 0.342 | 0.351 | 0.427 | 0.195 | 0.292 | 0.203 | 0.292 |

Table 1: Long-term forecasting results with different prediction lengths $T \in \{96, 192, 336, 720\}$. The best results are highlighted in bold.

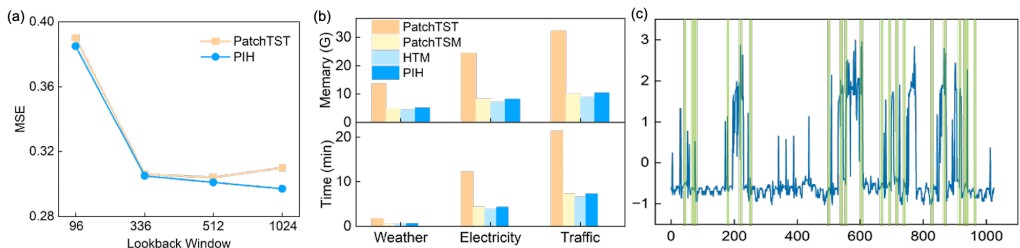

Figure 3: **(a)**: The performance comparison between PIH and PatchTST at $L \in \{96, 336, 512, 1024\}$. **(b)**: Comparison of GPU memory (GB) and training time (minutes/epoch) for PatchTST, PatchTST, HTM, and PIH. **(c)**: Visualization of a sample sequence in the *Electricity*, highlighting the most important 20 patches identified by the IBF module with green shading.

selected Mamba-based, Transformer-based, and Linear-based models as baselines. For Mamba-based model, we choose S-Mamba [34] and Bi-Mamba [12]. For Transformer-based models, in addition to PatchTST, we selected FEDformer [43], Autoformer [35], and Informer [42]. Furthermore, we include two Linear-based models, DLinear and NLinear [40]. Given that these two models were proposed to address the limitations of Transformer-based models in handling long lookback windows, we also set $L = 1024$ for them. All models follow the same experimental setup, with prediction lengths $T \in \{96, 192, 336, 720\}$. We use MSE and MAE as evaluation metrics. We conducted five repeated experiments and provided the **significance analysis** in the Appendix C.5.

**Results and Analysis.** As show in Table 1. For models like S-Mamba, Transformer, Autoformer, and Informer, PIH significantly outperforms them. Even for models specifically designed to handle long windows, such as PatchTST, DLinear, and NLinear, PIH still surpasses them, demonstrating its effectiveness in processing longer sequences. It is worth noting that we did not intentionally choose an unusual setting like $L = 1024$ to lower the performance of these three models. In Appendix C.1, we also provide their performance under shorter windows (e.g., 336 and 512), where PIH continues to outperform them. Overall, PIH with a much longer window setting achieves better results than other models with shorter windows. Our experiments highlight the potential for further increasing the window size.

| | Transformer | | | | Informer | | | | Autoformer | | | | PatchTST | | | |
|---|---|---|---|---|---|---|---|---|---|---|---|---|---|---|---|---|
| | Origin | | Ours | | Origin | | Ours | | Origin | | Ours | | Origin | | Ours | |
| | MEW | MSE | MEW | MSE | MEW | MSE | MEW | MSE | MEW | MSE | MEW | MSE | MEW | MSE | MEW | MSE |
| Etth1 | 72 | 1.009 | **144** | **0.96** | 72 | 1.146 | **144** | **1.063** | 72 | 0.508 | **192** | **0.491** | 512 | 0.447 | **1024** | **0.435** |
| Etth2 | 336 | 2.297 | **512** | **2.087** | 48 | 3.624 | **96** | **3.289** | 120 | 0.458 | **168** | **0.446** | 336 | 0.379 | **1024** | **0.378** |
| Weather | **512** | 0.504 | **512** | **0.482** | 48 | 1.028 | **120** | **0.957** | 144 | 0.460 | **228** | **0.451** | **1024** | **0.306** | **1024** | 0.309 |
| Traffic | 168 | 0.686 | **192** | **0.680** | 48 | 1.083 | **72** | **0.973** | 144 | 0.649 | **192** | **0.612** | 512 | 0.432 | **1024** | **0.430** |
| Ettm1 | 48 | 0.974 | **228** | **0.843** | 96 | 0.977 | **168** | **0.910** | 192 | 0.529 | **228** | **0.517** | 1024 | 0.407 | **1440** | **0.399** |
| Ettm2 | 120 | 2.784 | **168** | **2.436** | 96 | 3.956 | **168** | **3.648** | 336 | 0.419 | **336** | **0.410** | 1024 | 0.350 | **1024** | **0.342** |
| Electricity | 48 | 0.287 | **96** | **0.281** | 72 | 0.372 | **120** | **0.361** | 336 | 0.243 | **336** | **0.237** | 512 | 0.197 | **1024** | **0.192** |

Table 2: The MEW and MSE of the "origin" Transformer-based models: Transformer, Informer, Autoformer, and PatchTST, as well as the versions integrated with our module ("ours"). Larger MEWs and smaller MSEs are highlighted in bold.

**PIH Can Effectively Utilize Longer Lookback Windows.** As shown in Fig. 3 (a), we set the lookback window to $L = \{96, 336, 512, 1024\}$ and used the average MSE over 7 datasets with forecasting horizons of $T \in \{96, 192, 336, 720\}$ as the evaluation metric. The results indicate that the performance of PatchTST improves steadily as the window increases from 96 to 512, but declines when extended to 1024. In contrast, PIH exhibits a consistent performance improvement as the window size increases from 96 to 1024. This suggests that the HTM and IBF modules help PatchTST improve MEW, leading to better performance with longer windows. Another noteworthy observation is that, except for $L = 96$, PIH consistently outperforms PatchTST for the same $L$. We hypothesize that with $L = 96$, sequence redundancy and noise is low, and the patch strategy alone is sufficient to manage it effectively, rendering IBF and HTM unnecessary. Consequently, PIH lags behind PatchTST at this window size. However, as the window length increases and sequence redundancy grows, the IBF and HTM modules become more effective, allowing PIH to surpass PatchTST.

**Computational Overhead.** In addition to performance comparisons, we evaluated computation time and memory usage, as shown in Fig. 3 (b). When using only the HTM module without the IBF (referred to as HTM), it demonstrates significant improvements in both computational time and memory usage compared to the pure Transformer architecture (referred to as PatchTST), surpassing it by a notable margin (2 to 3 times). Additionally, HTM outperforms the pure Mamba architecture (referred to as PatchTSM), which can be attributed to the Transformer's lower computational cost when handling shorter sequences compared to Mamba (see Appendix E.2). Moreover, when both HTM and IBF are integrated (i.e., PIH), the additional overhead introduced is negligible, as the IBF module only consists of a simple MLP.

**Interpretability of IBF.** Another advantage of incorporating the IBF module is its ability to enhance interpretability by identifying crucial subsequences for the final prediction. As shown in Fig. 3 (c), we provide a visualization of a sample from the *Electricity* dataset. Based on Eq. 5, we select the top 20 most important patches and highlight them in green. The results indicate that the model focuses more on sequences at peak positions. This aligns with prior knowledge, as peak positions typically represent changes in the sequence and contain more critical information.

## 4.2 The Generalizability of Our Modules

We conducted experiments on three different point-based Transformer models—Transformer, Informer, and Autoformer—as well as a patch-based PatchTST. Specifically, for Transformer, Informer, and Autoformer, we selected MEW from the following windows: 48, 72, 96, 120, 144, 168, 192, 228, 336, and 512. For PatchTST, the MEW were selected from the set: 72, 96, 120, 144, 168, 192, 228, 336, 512, 1024, and 1440. The prediction length was set to $T = 720$, with MSE used as the evaluation metric for MEW. All other experimental settings remained the same. The default model and the versions with the two modules integrated are denoted as "origin" and "Ours", respectively. As shown in Table 2, the MEW of point-based models tends to be quite small, especially for Informer, where the MEW on all 7 datasets does not exceed 96. On the other hand, the PatchTST, which is patch-based, exhibits larger MEW. This is because the use of patches effectively reduces the noise in the lookback window and significantly reduces the number of tokens, alleviating the issues of attention dispersion and overfitting to noise in the Transformer. After integrating our modules, the MEW of all four models showed significant improvement, and they also achieved better MSE. This is because the IBF module effectively reduces the noise in the input, while the HTM further diminishes noise and, by decomposing long token sequences into shorter ones, ensures that the Transformer

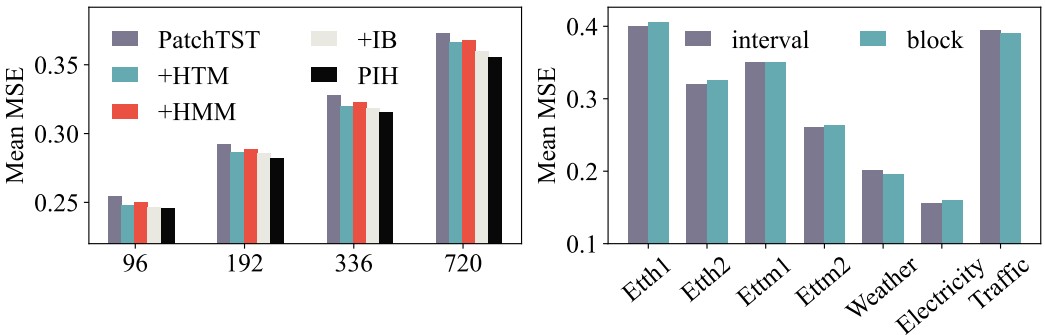

Figure 4: Left: Ablation experiments of different modules at prediction lengths $T = \{96, 192, 336, 720\}$. Right: Comparison of *interval split* and *block split*. The average MSE across 7 datasets at $T = \{96, 192, 336, 720\}$ is used as the evaluation metric.

processes a noise-filtered, shorter sequence. These experiments demonstrate the generalizability of our modules. Furthermore, we observe that for the same model, when the MEW is improved, its performance also increases, indicating that enhancing a model's MEW is a promising direction.

### 4.3 Ablation Study

**Component Ablation.** To assess the effectiveness of IBF and HTM, we utilize PatchTST as a baseline, upon which we separately introduce IBF, HTM and both simultaneously to obtain three variants: +IB, +HTM, and PIH. Additionally, we introduce a variant of +HTM, +HMM, which solely employs Mamba to handle both the original long sequences and the divided short sequences. All experiments maintain consistent settings, with $L = 1024$ and $T = \{96, 192, 336, 720\}$. The average MSE across 7 datasets is used as the evaluation metric. As illustrated in Fig. 4, the following observations are made: (1) Both IBF and HTM modules enhance the model's performance, and combining these two modules yields superior results. (2) Compared to HMM, HTM exhibits slightly better performance, which can be attributed to the different mechanisms between Transformer and Mamba, making each more suited to handling different types of sequences. By combining the strengths of both, the hybrid approach achieves superior results. As discussed earlier, the Transformer has lower computational costs for shorter sequences, while Mamba is more efficient for longer sequences. Therefore, from both performance and computational overhead perspectives, using a combination of both architectures is a better choice than relying solely on one. (3) At longer prediction lengths, such as $T = 720$, our model demonstrates greater improvements compared to $T = 96$, indicating that larger windows $L$ provide more significant benefits for longer-term predictions (longer $T$).

**Interval Split vs. Block Split.** We compared the performance of *interval split* and *block split* across various datasets, as shown in Fig. 4. Overall, the effectiveness of both split methods is roughly comparable, demonstrating their ability to preserve sequential characteristics. However, slight variations in performance are observed across different datasets. We speculate that this discrepancy arises from the different strengths of each partitioning method in retaining specific sequential patterns. Intuitively, *interval split* emphasizes global variations, while *block split* focuses on variations within specific periods. Determining the most suitable partitioning strategy remains a topic for future exploration.

**Other Hyperparameters.** Our model incorporates several crucial hyperparameters, including $K$, determining the number of partitions; $\beta$, which governs the balance between prediction and compression in the IB objective; and the temperature factor $\tau$, influencing subsequence sampling. We investigate the impact of $K \in \{2, 4, 6, 8\}$, $\beta \in \{0.0001, 0.001, 0.1, 1\}$, and set $\tau \in \{0.1, 0.5, 1, 2\}$. We find that the choice of $K$ does not significantly affect performance, whereas $\tau$ and $\beta$ exhibit considerable influence on performance, likely due to variations in the redundancy levels across different datasets. Detailed hyperparameter experiments can be found in the Appendix C.4.

## 5 Conclusion

In this paper, we propose a MEW metric to evaluate the model's ability to leverage the lookback window. We introduce two model-agnostic modules, IBF and HTM, from both information-theoretic and model-architectural perspectives. Experiments show that these modules can effectively improve the model's MEW, and with a larger MEW, the model's performance also improves, demonstrating the importance of enhancing MEW. Furthermore, we combine these modules with the patch strategy to design the PIH model, which can handle longer windows than previous works and achieves state-of-the-art results, illustrating the potential of leveraging longer windows.

## 6 Acknowledgement

This paper is partially supported by the National Natural Science Foundation of China (No.12227901). The AI-driven experiments, simulations and model training were performed on the robotic AI-Scientist platform of Chinese Academy of Sciences., Anhui Science Foundation for Distinguished Young Scholars (No.1908085J24), Natural Science Foundation of China (No.62502491).

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

# A  Limitations

First, our experiments demonstrate that extending the window length to $L = 1024$ still yields performance improvements, suggesting that further exploration of longer windows is a promising direction. Secondly, although the modules we propose effectively improve the model's MEW, a bottleneck still exists. A more promising avenue of research is to explore whether there is a scaling law for lookback windows, meaning that larger windows consistently result in better performance. Thirdly, the *interval split* and *block split* methods are heuristic. Designing an adaptive, end-to-end split method tailored to each training dataset may lead to better results. Lastly, while recent large time-series models have adopted much longer windows, we assert that our approach is orthogonal to theirs. It is worth exploring whether our method can be integrated into these large models (e.g.,LLM-based) to further extend their window sizes.

# B  Dataset Description

In the main text, we use seven popular multivariate datasets provided in [35] for forecasting and representation learning. The Weather dataset collects 21 meteorological indicators in Germany, such as humidity and air temperature. The Traffic dataset records road occupancy rates from various sensors on San Francisco freeways. The Electricity dataset describes the hourly electricity consumption of 321 customers. The ETT (Electricity Transformer Temperature) datasets are collected from two different electric transformers labeled as 1 and 2, each containing two resolutions (15 minutes and 1 hour), denoted as m and h, respectively. Thus, there are four ETT datasets in total: ETTm1, ETTm2, ETTh1, and ETTh2. In addition, we incorporated two additional datasets: Solar[10] and PeMS[16]. Their experimental results are shown in Table 4. The results indicate that PIH also achieved the best performance on these two datasets.

Table 3: Statistics of popular datasets for benchmark.

| Datasets | Weather | Traffic | Electricity | ETTh1 | ETTh2 | ETTm1 | ETTm2 | Solar | PEMS |
|---|---|---|---|---|---|---|---|---|---|
| Features | 21 | 862 | 321 | 7 | 7 | 7 | 7 | 137 | 358 |
| Timesteps | 52696 | 17544 | 26304 | 17420 | 17420 | 69680 | 69680 | 52,179 | 21,351 |

# C  Experiments

## C.1  Experiments settings

PIH is built upon the PatchTST and thus incorporates all hyperparameters from PatchTST. To ensure a fair comparison, we adhered strictly to the settings of PatchTST for these shared hyperparameters, with the exception of the learning rate. We conducted a hyperparameter search only for those introduced by the HTM and IBF modules, as this was necessary. The only exception is the learning rate. Given the introduction of the Mamba and IBF modules, the default learning rate of $lr = 0.0001$ in PatchTST is suboptimal. Consequently, we set the search space for the PIH learning rate to $lr = \{0.001, 0.0005, 0.0001\}$. To ensure a fair comparison, we also performed a hyperparameter search for the learning rate in PatchTST, and selected the optimal results. The resulting mean Absolute Error (MAE) values were 0.310 and 0.335, which are almost unchanged compared to the default learning rate ($lr = 0.0001$), yielding 0.310 and 0.336. Thus, this does not affect our result analysis.

Our model incorporates several crucial hyperparameters, including $K$, which determines the number of partitions; $\beta$, which governs the balance between prediction and compression in the information bottleneck (IB) objective; and the temperature factor $\tau$, which influences subsequence sampling. We set $K \in \{2, 4\}$, $\beta \in \{0.0001, 0.001, 0.1, 1\}$, and $\tau \in \{0.1, 0.5, 1, 2\}$. We selected the optimal hyperparameters based on the results from the validation set.

The original PatchTST paper discusses the impact of key hyperparameters in detail, such as the number of Transformer layers, patch length, the number of heads in multi-head attention, and the dimension of the latent space. The default parameters provided in their official code represent the best-performing combination overall, so there was no need to repeat hyperparameter searches for PatchTST. Additionally, since our PIH is based on PatchTST, any changes to PatchTST would necessitate corresponding adjustments to PIH. Conducting performance comparisons between PIH

and PatchTST for every parameter setting would render the ablation experiments overly redundant. Therefore, we did not adjust the hyperparameters of PatchTST. The only exception is the learning rate. Given the introduction of the Mamba and IBF modules, the default learning rate of $lr = 0.0001$ in PatchTST is suboptimal. Consequently, we set the search space for the PIH learning rate to $lr = \{0.001, 0.0005, 0.0001\}$. To ensure a fair comparison, we also performed a hyperparameter search for the learning rate in PatchTST, using $lr = \{0.001, 0.0005, 0.0001\}$, and selected the optimal results. The resulting mean Absolute Error (MAE) values were 0.310 and 0.335, which are almost unchanged compared to the default learning rate ($lr = 0.0001$), yielding 0.310 and 0.336. Thus, this does not affect our result analysis.

## C.2 Additional Experimental Results.

Due to space limitations in the main text, we only discussed the results on seven datasets. To better evaluate our model, we also conducted experiments on the Solar and PEMS datasets, with the experimental settings consistent with those described in the main text. As shown in Table 4, our model also outperforms other methods on these two datasets.

Table 4: Multivariate long-term series forecasting results for the Solar and PEMS.

| Models | | PIH | | PatchTST | | iTransformer | | DLinear | | MICN | | FourierGNN | | FEDformer | | Autoformer | |
|---|---|---|---|---|---|---|---|---|---|---|---|---|---|---|---|---|---|---|
| Metric | | MSE | MAE | MSE | MAE | MSE | MAE | MSE | MAE | MSE | MAE | MSE | MAE | MSE | MAE | MSE | MAE |
| Solar | 96 | **0.163** | **0.230** | 0.185 | 0.246 | 0.170 | 0.246 | 0.191 | 0.257 | 0.190 | 0.243 | 0.183 | 0.232 | 0.214 | 0.311 | 0.316 | 0.369 |
| | 192 | **0.177** | **0.239** | 0.201 | 0.262 | 0.195 | 0.263 | 0.211 | 0.273 | 0.205 | 0.247 | 0.198 | 0.256 | 0.281 | 0.364 | 0.418 | 0.437 |
| | 336 | **0.188** | **0.247** | 0.209 | 0.266 | 0.217 | 0.282 | 0.228 | 0.285 | 0.219 | 0.250 | 0.205 | 0.261 | 0.294 | 0.378 | 0.438 | 0.467 |
| | 720 | **0.196** | **0.255** | 0.226 | 0.283 | 0.208 | 0.276 | 0.236 | 0.294 | 0.227 | 0.263 | 0.202 | 0.265 | 0.315 | 0.406 | 0.618 | 0.550 |
| PEMS | 12 | **0.060** | **0.163** | 0.063 | 0.166 | 0.064 | 0.167 | 0.078 | 0.187 | 0.094 | 0.204 | 0.091 | 0.202 | 0.283 | 0.394 | 0.584 | 0.607 |
| | 24 | **0.075** | **0.179** | 0.080 | 0.185 | 0.081 | 0.187 | 0.113 | 0.224 | 0.116 | 0.229 | 0.116 | 0.232 | 0.300 | 0.431 | 0.672 | 0.664 |
| | 48 | **0.100** | **0.204** | 0.109 | 0.213 | 0.111 | 0.215 | 0.167 | 0.274 | 0.147 | 0.255 | 0.165 | 0.271 | 0.396 | 0.476 | 0.879 | 0.781 |
| | 96 | **0.132** | **0.233** | 0.145 | 0.243 | 0.142 | 0.240 | 0.212 | 0.313 | 0.256 | 0.362 | 0.196 | 0.300 | 0.477 | 0.537 | 1.100 | 0.895 |

## C.3 Performance of PatchTST, DLinear, and NLinear under Different Window Lengths

In the main text, we set $L$ to 1024. This is not done with the intention of deliberately undermining the performance of PatchTST, DLinear, and NLinear under an unusual setting. Here, we conducted experiments with DLinear and NLinear, two linear-based models, under two settings: $L = 336$ and $L = 1024$, with the results shown in Table 5. For PatchTST, we set $L = 336$ and $L = 512$. As shown in Table 5, even with the shorter window settings, their performance still lags behind that of PIH at $L = 1024$. Specifically, we can draw the following conclusions:

- Linear-based models indeed perform well against noise, with NLinear(1024) generally outperforming NLinear(336). This is consistent with the results of PIH, indicating that larger windows are beneficial.

- NLinear(1024) generally outperforms NLinear(336), whereas DLinear(1024) underperforms compared to DLinear(336). Thus, directly increasing the window size in linear-based methods is not always effective.

- PIH(1024) outperforms NLinear(1024), which can be attributed to the superior representational capabilities of the Transformer and Mamba modules compared to linear modules. Therefore, it is essential to continue exploring the potential of Transformer-based models with longer windows rather than relying solely on linear-based models.

- Compared to PatchTST (336), PatchTST (512), and PatchTST (1024) discussed in the main text, PIH (1024) achieves better performance. This can be attributed to the IBF and HTM modules, which enable PIH to leverage larger windows effectively.

## C.4 Hyperparameters

Our model incorporates several crucial hyperparameters, including $K$, determining the number of partitions; $\beta$, which governs the balance between prediction and compression in the IB objective; and the temperature factor $\tau$, influencing subsequence sampling. We investigate the impact of $K \in \{2, 4, 6, 8\}$, $\beta \in \{0.0001, 0.001, 0.1, 1\}$, and set $\tau \in \{0.1, 0.5, 1, 2\}$. As shown in Fig. 5, we investigate the effects of the three hyperparameters on the ETTh1, ETTm1, and Weather datasets.

Table 5: Comparison between DLinear, NLinear, PatchTST and PIH with different lookback windows.

| | | Weather | | Traffic | | Electricity | | ETTh1 | | ETTh2 | | ETTm1 | | ETTm2 | |
|---|---|---|---|---|---|---|---|---|---|---|---|---|---|---|---|
| | | MSE | MAE | MSE | MAE | MSE | MAE | MSE | MAE | MSE | MAE | MSE | MAE | MSE | MAE |
| DLinear(336) | 96 | 0.176 | 0.237 | 0.410 | 0.282 | 0.140 | 0.237 | 0.375 | 0.399 | 0.289 | 0.353 | 0.299 | 0.343 | 0.167 | 0.260 |
| | 192 | 0.220 | 0.282 | 0.423 | 0.287 | 0.153 | 0.249 | 0.405 | 0.416 | 0.383 | 0.418 | 0.335 | 0.365 | 0.224 | 0.303 |
| | 336 | 0.265 | 0.319 | 0.436 | 0.296 | 0.169 | 0.267 | 0.439 | 0.443 | 0.448 | 0.465 | 0.369 | 0.386 | 0.281 | 0.342 |
| | 720 | 0.323 | 0.362 | 0.466 | 0.315 | 0.203 | 0.301 | 0.472 | 0.490 | 0.605 | 0.551 | 0.425 | 0.421 | 0.397 | 0.421 |
| DLinear(1024) | 96 | 0.167 | 0.225 | 0.385 | 0.275 | 0.132 | 0.229 | 0.378 | 0.403 | 0.294 | 0.361 | 0.314 | 0.358 | 0.164 | 0.260 |
| | 192 | 0.211 | 0.267 | 0.397 | 0.279 | 0.146 | 0.243 | 0.414 | 0.428 | 0.430 | 0.448 | 0.356 | 0.391 | 0.238 | 0.317 |
| | 336 | 0.255 | 0.304 | 0.412 | 0.288 | 0.161 | 0.260 | 0.453 | 0.458 | 0.492 | 0.484 | 0.365 | 0.388 | 0.265 | 0.326 |
| | 720 | 0.313 | 0.351 | 0.450 | 0.309 | 0.195 | 0.292 | 0.511 | 0.520 | 0.905 | 0.683 | 0.410 | 0.417 | 0.338 | 0.375 |
| NLinear(336) | 96 | 0.182 | 0.232 | 0.410 | 0.279 | 0.141 | 0.237 | 0.374 | 0.394 | 0.277 | 0.338 | 0.306 | 0.348 | 0.167 | 0.255 |
| | 192 | 0.225 | 0.269 | 0.410 | 0.279 | 0.154 | 0.248 | 0.408 | 0.415 | 0.344 | 0.381 | 0.349 | 0.375 | 0.221 | 0.293 |
| | 336 | 0.271 | 0.301 | 0.435 | 0.290 | 0.171 | 0.265 | 0.429 | 0.427 | 0.357 | 0.400 | 0.375 | 0.388 | 0.274 | 0.327 |
| | 720 | 0.338 | 0.348 | 0.464 | 0.307 | 0.210 | 0.297 | 0.440 | 0.453 | 0.394 | 0.436 | 0.433 | 0.422 | 0.368 | 0.384 |
| NLinear(1024) | 96 | 0.170 | 0.226 | 0.383 | 0.270 | 0.133 | 0.229 | 0.379 | 0.404 | 0.296 | 0.351 | 0.317 | 0.359 | 0.163 | 0.257 |
| | 192 | 0.215 | 0.265 | 0.397 | 0.274 | 0.148 | 0.242 | 0.414 | 0.426 | 0.337 | 0.382 | 0.352 | 0.381 | 0.216 | 0.294 |
| | 336 | 0.259 | 0.298 | 0.410 | 0.281 | 0.164 | 0.259 | 0.442 | 0.445 | 0.359 | 0.407 | 0.374 | 0.393 | 0.265 | 0.326 |
| | 720 | 0.321 | 0.342 | 0.449 | 0.303 | 0.203 | 0.292 | 0.470 | 0.477 | 0.417 | 0.456 | 0.409 | 0.413 | 0.338 | 0.375 |
| PatchTST(336) | 96 | 0.152 | 0.199 | 0.367 | 0.251 | 0.130 | 0.222 | 0.375 | 0.399 | 0.274 | 0.336 | 0.290 | 0.342 | 0.165 | 0.255 |
| | 192 | 0.197 | 0.243 | 0.385 | 0.259 | 0.148 | 0.240 | 0.414 | 0.421 | 0.339 | 0.379 | 0.332 | 0.369 | 0.220 | 0.292 |
| | 336 | 0.249 | 0.283 | 0.398 | 0.265 | 0.167 | 0.261 | 0.431 | 0.436 | 0.331 | 0.380 | 0.366 | 0.392 | 0.278 | 0.329 |
| | 720 | 0.320 | 0.335 | 0.434 | 0.287 | 0.202 | 0.291 | 0.449 | 0.466 | 0.379 | 0.422 | 0.420 | 0.424 | 0.367 | 0.385 |
| PatchTST(512) | 96 | 0.149 | 0.198 | 0.360 | 0.249 | 0.129 | 0.222 | 0.370 | 0.400 | 0.274 | 0.337 | 0.293 | 0.346 | 0.166 | 0.256 |
| | 192 | 0.194 | 0.241 | 0.379 | 0.256 | 0.147 | 0.240 | 0.413 | 0.429 | 0.341 | 0.382 | 0.333 | 0.370 | 0.223 | 0.296 |
| | 336 | 0.245 | 0.282 | 0.392 | 0.264 | 0.163 | 0.259 | 0.422 | 0.440 | 0.329 | 0.384 | 0.369 | 0.392 | 0.274 | 0.329 |
| | 720 | 0.314 | 0.334 | 0.432 | 0.286 | 0.197 | 0.290 | 0.447 | 0.468 | 0.379 | 0.422 | 0.416 | 0.420 | 0.362 | 0.385 |
| PIH(1024) | 96 | **0.147** | **0.198** | **0.357** | **0.248** | **0.127** | **0.220** | **0.360** | **0.394** | **0.263** | **0.328** | **0.291** | **0.349** | **0.161** | **0.253** |
| | 192 | **0.191** | **0.239** | **0.371** | **0.255** | **0.145** | **0.240** | **0.396** | **0.418** | **0.324** | **0.370** | **0.337** | **0.374** | **0.213** | **0.289** |
| | 336 | **0.241** | **0.280** | **0.392** | **0.261** | **0.160** | **0.256** | **0.409** | **0.432** | **0.314** | **0.376** | **0.360** | **0.386** | **0.265** | **0.326** |
| | 720 | **0.309** | **0.329** | **0.430** | **0.282** | **0.192** | **0.287** | **0.435** | **0.466** | **0.378** | **0.425** | **0.405** | **0.411** | **0.342** | **0.375** |

We find that the choice of $K$ does not significantly affect performance, whereas $\tau$ and $\beta$ exhibit considerable influence, likely due to variations in the redundancy levels across different datasets.

## C.5   Robustness of Our Results

To verify whether the improvements of PIH over PatchTST are statistically significant, we utilized p-values to evaluate the prediction results. Specifically, for a prediction length of $T = 96$, we validated the significance of the improvement achieved by PIH over PatchTST. Table 6 presents the p-values from 5 experiments conducted at $T = 96, 192, 336, 720$. The results demonstrating that the performance improvements are significant in 5 out of 7 datasets. (p-value $< 0.05$)

Table 6: P-values for the significance of PIH improvements over PatchTST at different prediction lengths.

| $T$ | Weather | Traffic | Electricity | Etth1 | Etth2 | Ettm1 | Ettm2 |
|---|---|---|---|---|---|---|---|
| 96 | 0.397 | $1.19 \times 10^{-6}$ | $7.00 \times 10^{-5}$ | $1.10 \times 10^{-5}$ | $1.99 \times 10^{-6}$ | 0.32 | 0.002 |
| 192 | 0.49 | $7.47 \times 10^{-9}$ | $7.78 \times 10^{-5}$ | $1.60 \times 10^{-6}$ | $8.65 \times 10^{-6}$ | 0.20 | 0.013 |
| 336 | 0.015 | $6.40 \times 10^{-7}$ | $4.60 \times 10^{-4}$ | $2.30 \times 10^{-6}$ | $2.44 \times 10^{-7}$ | 0.008 | 0.007 |
| 720 | 0.08 | $6.29 \times 10^{-6}$ | $2.92 \times 10^{-6}$ | $2.53 \times 10^{-6}$ | $2.05 \times 10^{-7}$ | 0.36 | 0.0008 |

## C.6   Integration into the iTransformer

iTransformer [18] is a special Transformer-based time series model that, instead of treating points or patches as tokens, treats each channel as a token. Therefore, the number of input tokens for iTransformer is determined by the number of channels rather than the length of the lookback windows. Although this differs slightly from the main focus of this paper, we found that IBF and HTM can also be applied to iTransformer. In this case, the goal of these two modules becomes reducing redundancy among channels and decreasing the number of tokens fed into the Transformer to mitigate the attention dispersion problem. We conducted experiments on datasets with a large number of

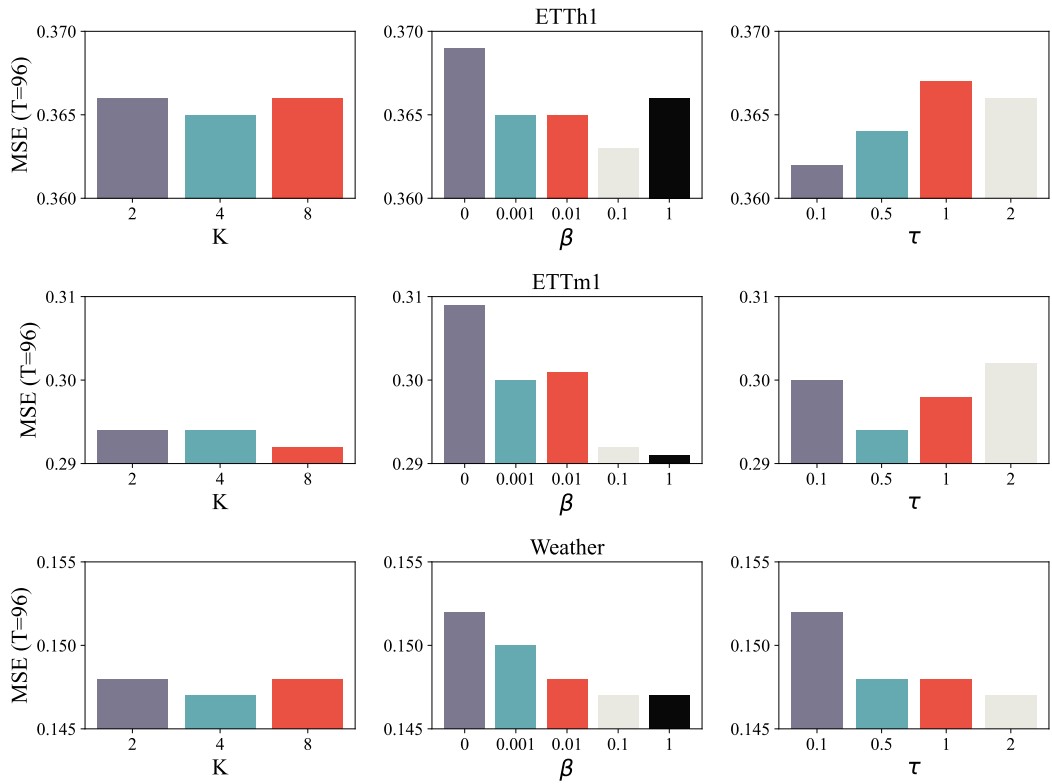

Figure 5: The impact of hyperparameters $K$, $\beta$, and $\tau$ in the PIH model on performance across ETTh1, ETThm1, and Weather datasets. The vertical axis represents the MSE at $T = 96$.

Table 7: Comparison of MSE values for different datasets and horizons between our method and iTransformer.

| Dataset | | Ours | | iTransformer | |
|---|---|---|---|---|---|
| | | MSE | MAE | MSE | MAE |
| Electricity | 96 | **0.129** | **0.225** | 0.133 | 0.229 |
| | 192 | **0.146** | **0.242** | 0.155 | 0.251 |
| | 336 | **0.163** | **0.260** | 0.167 | 0.264 |
| | 720 | **0.189** | **0.285** | 0.194 | 0.288 |
| Traffic | 96 | **0.346** | **0.252** | 0.349 | 0.255 |
| | 192 | **0.354** | **0.254** | 0.359 | 0.263 |
| | 336 | **0.370** | **0.261** | 0.379 | 0.272 |
| | 720 | **0.413** | **0.280** | 0.417 | 0.291 |
| PEMS | 12 | **0.059** | **0.158** | 0.064 | 0.167 |
| | 24 | **0.075** | **0.178** | 0.081 | 0.187 |
| | 48 | **0.101** | **0.203** | 0.111 | 0.215 |
| | 96 | **0.133** | **0.234** | 0.142 | 0.240 |

channels, including Electricity, Traffic, and PEMS. As shown in Table C.6, after integrating our module, the performance of iTransformer is further enhanced.

## D  Proofs of IB

### D.1  Proof of Eq(7)

We first examine the first term $-I\left(\mathbf{z}^{\text{noise}}, \mathbf{Y}\right)$ which encourages $\mathbf{z}_{\text{noise}}$ is informative of label $\mathbf{Y}$ .

$$-I\left(\mathbf{z}^{\text{noise}}, \mathbf{Y}\right) \leq \mathbb{E}_{\mathbf{Y}, \mathbf{z}^{\text{noise}}} - \log q_\theta\left(\mathbf{Y} \mid \mathbf{z}^{\text{noise}}\right)$$
$$:= \mathcal{L}_{\text{pred}}\left(\mathbf{z}^{\text{noise}}, Y\right) \tag{13}$$

Here, $p_\theta\left(\mathbf{Y} \mid \mathbf{z}^{\text{noise}}\right)$ represents the variational approximation to the true posterior distribution $p\left(\mathbf{Y} \mid \mathbf{z}^{\text{noise}}\right)$.This equation illustrates that minimizing $-I\left(\mathbf{z}^{\text{noise}}, \mathbf{Y}\right)$ is achieved by minimizing the prediction loss between $\mathbf{z}^{\text{noise}}$ and $\mathbf{Y}$. We choose to utilize the Mean Squared Error (MSE) loss to quantify the disparity between the prediction and the ground truth.

Here we provide more details about how to yield Eq. 13. By the definition of mutual information and introducing variational approximation $p_\theta\left(\mathbf{Y} \mid \mathbf{z}^{\text{noise}}\right)$ of intractable distribution $p\left(\mathbf{Y} \mid \mathbf{z}^{\text{noise}}\right)$ , we have:

$$\begin{aligned} I\left(\mathbf{Y}, \mathbf{z}^{\text{noise}}\right) &= \mathbb{E}_{\mathbf{Y}, \mathbf{z}^{\text{noise}}}\left[\log \frac{p\left(\mathbf{Y} \mid \mathbf{z}^{\text{noise}}\right)}{p(\mathbf{Y})}\right] \\ &= \mathbb{E}_{\mathbf{Y}, \mathbf{z}^{\text{noise}}}\left[\log \frac{p_\theta\left(\mathbf{Y} \mid \mathbf{z}^{\text{noise}}\right)}{p(\mathbf{Y})}\right] \\ &\quad + \mathbb{E}_{\mathbf{z}^{\text{noise}}}\left[KL\left(p\left(\mathbf{Y} \mid \mathbf{z}^{\text{noise}}\right) \| p_\theta\left(\mathbf{Y} \mid \mathbf{z}^{\text{noise}}\right)\right)\right] \end{aligned} \tag{14}$$

According to the non-negativity of the KL divergence, we have:

$$\begin{aligned} I\left(\mathbf{Y}; \mathbf{z}^{\text{noise}}\right) &\geq \mathbb{E}_{\mathbf{Y}, \mathbf{z}^{\text{noise}}}\left[\log \frac{p_\theta\left(\mathbf{Y} \mid \mathbf{z}^{\text{noise}}\right)}{p(\mathbf{Y})}\right] \\ &= \mathbb{E}_{\mathbf{Y}, \mathbf{z}^{\text{noise}}}\left[\log p_\theta\left(\mathbf{Y} \mid \mathbf{z}^{\text{noise}}\right)\right] + H(\mathbf{Y}) \end{aligned}$$

We can ignore $H(\mathbf{Y})$ since it can be treated as a constant. We model $p_\theta\left(\mathbf{Y} \mid \mathbf{z}^{\text{noise}}\right)$ as a predictor parameterized by $\theta$, which generates the model prediction $\mathbf{Y}$ based on the input $\mathbf{z}^{\text{noise}}$. Thus, minimizing the upper bound of $-I\left(\mathbf{z}^{\text{noise}}, \mathbf{Y}\right)$ entails minimizing the model prediction loss $\mathcal{L}_{\text{pred}}\left(\mathbf{z}^{\text{noise}}, \mathbf{Y}\right)$. We opt to employ the Mean Squared Error (MSE) loss to quantify the difference between the prediction and the ground truth.

### D.2  Proof of Eq(9)

We derive the upper bound of $I\left(\mathbf{z}^{\text{noise}}, \mathbf{z}\right)$ by introducing the variation approximation $q\left(\mathbf{z}^{\text{noise}}\right)$ of distribution $p\left(\mathbf{z}^{\text{noise}}\right)$ :

$$\begin{aligned} I\left(\mathbf{z}^{\text{noise}}, \mathbf{z}\right) &= \mathbb{E}_{\mathbf{z}, \mathbf{z}^{\text{noise}}}\left[\log \frac{p_\phi\left(\mathbf{z}^{\text{noise}} \mid \mathbf{z}\right)}{p(\mathbf{z})}\right] \\ &= \mathbb{E}_{\mathbf{z}, \mathbf{z}^{\text{noise}}}\left[\log \frac{p_\phi\left(\mathbf{z}^{\text{noise}} \mid \mathbf{z}\right)}{q(\mathbf{z}^{\text{noise}})}\right] \\ &\quad - \mathbb{E}_{\mathbf{z}^{\text{noise}}, \mathbf{z}}\left[KL\left(p\left(\mathbf{z}^{\text{noise}}\right) \| q\left(\mathbf{z}^{\text{noise}}\right)\right)\right] \end{aligned} \tag{15}$$

According to the non-negativity of KL divergence, we have:

$$I\left(\mathbf{z}^{\text{noise}}, \mathbf{z}\right) \leq \mathbb{E}_{\mathbf{z}}\left[KL\left(p_\phi\left((\mathbf{z}^{\text{noise}} \mid \mathbf{z}\right) \| q\left(\mathbf{z}^{\text{noise}}\right)\right)\right] \tag{16}$$

we assume that $q\left(\mathbf{z}^{\text{noise}}\right)$ is obtained by aggregating the patch representations in a fully perturbed sequences. The noise $\epsilon \sim \mathcal{N}\left(\mu_{\mathbf{z}}, \sigma_{\mathbf{z}}^2\right)$ is sampled from a Gaussian distribution where $\mu_{\mathbf{z}}$ and $\sigma_{\mathbf{z}}^2$ are mean and variance of $\mathbf{z}$. Choosing sum pooling as the aggregatiion function, since the summation of Gaussian distributions is a Gaussian, we have the following equation:

$$q\left(\mathbf{z}^{\text{noise}}\right) = \mathcal{N}\left(N\mu_{\mathbf{z}}, N\sigma_{\mathbf{z}}^2\right) \tag{17}$$

Then for $p_\phi \left( \mathbf{z}^{\text{noise}} \mid \mathbf{z} \right)$, we have the following equation:

$$p_\phi \left( \mathbf{z}^{\text{noise}} \mid \mathbf{z} \right) = \mathcal{N} \left( N\mu_{\mathbf{z}} + \sum_{j=1}^{N} \lambda_j \mathbf{z}_j - \sum_{j=1}^{N} \lambda_j \mu_{\mathbf{z}}, \sum_{j=1}^{N} \left( 1 - \lambda_j \right)^2 \sigma_{\mathbf{z}}^2 \right) \tag{18}$$

Finally, we have following inequality by plugging Eq. 17 and Eq. 18 into Eq. 16:

$$I \left( \mathbf{z}^{\text{noise}}, \mathbf{z} \right) \leq \mathbb{E}_{\mathbf{z}} \left[ -\frac{1}{2} \log A + \frac{1}{2N} A + \frac{1}{2N} B^2 \right] + C \tag{19}$$

where $A = \sum_{j=1}^{N} \left( 1 - \lambda_j \right)^2$, $B = \frac{\sum_{j=1}^{N} \lambda_j \left( \mathbf{z}_j - \mu_{\mathbf{z}} \right)}{\sigma_{\mathbf{H}^1}}$ and $C$ is a constant term which is ignored during optimization.

## E  Others

### E.1  Relationship with Large Time-Series Models

Although some recent large time-series models are capable of handling longer windows, they rely on significantly more parameters and much larger training datasets compared to our experiments. Additionally, when tested on the same datasets we used, these models still employ smaller window sizes. Our work does not conflict with these advancements in large time-series models. This is because the HTM and IBF modules we propose are model-agnostic and can be integrated into large time-series models, a direction worth exploring in future.

### E.2  Mamba vs Transformer

We analyze HTM from both performance and computational overhead perspectives and find that the hybrid architecture has distinct advantages over using only Mamba or Transformer.

**From a performance perspective.**    The ablation experiments presented in Fig.3(b) of main paper indicate that removing the Transformer results in slightly worse performance, highlighting the significant advantage of the combined Transformer and Mamba architecture. This finding is further supported by recent works such as Mamba-2-Hybrid [32], Dimba [6], and Jamba [13].

Table 8: Comparison of GPU memory usage and training time per epoch for a single-layer Transformer and Mamba on the Weather dataset as the lookback window $L$ varies.

| Model | Metric | 96 | 192 | 336 | 512 | 1024 |
|---|---|---|---|---|---|---|
| Mamba | Time (s) | 18.76 | 21.63 | 28.25 | 36.52 | 58.47 |
| | Memory (G) | 2.02 | 3.30 | 4.90 | 6.78 | 9.53 |
| Transformer | Time (s) | 7.33 | 17.94 | 27.84 | 44.70 | 96.57 |
| | Memory (G) | 0.75 | 1.64 | 3.21 | 5.56 | 15.05 |

**Considering computational overhead.**    Our framework employs the Transformer solely to process the partitioned short subsequences, which generally mitigates concerns about the costs associated with the Transformer. To validate this, we compared the computation time and GPU memory usage between using a single layer of Mamba and a single layer of Transformer under various lookback window settings (with nearly identical parameter counts). As shown in Table 8, when $L \leq 336$, the computational overhead of the Transformer is even lower than that of Mamba; however, at $L = 1024$, the computational cost of the Transformer is nearly twice that of Mamba. In our experiments, $K$ is typically set to 4, resulting in a subsequence length of $L/K = 1024/4 < 336$. Consequently, the addition of the Transformer module incurs less overhead compared to using only Mamba.

In summary, we conclude that retaining the Transformer module is essential for enhancing performance while managing computational costs effectively.

