# OpenReview forum: "Enhancing the Maximum Effective Window for Long-Term Time Series Forecasting"
_NeurIPS.cc/2025/Conference — NeurIPS 2025 poster_

### Official Review · Reviewer_up9s · 2025-06-29

**Clarity:** 2
**Significance:** 2
**Originality:** 1
**Rating:** 3
**Confidence:** 4

**Summary:**

This paper introduces the Maximum Effective Window (MEW) and proposes two model-agnostic modules: Information Bottleneck Filter (IBF) for reducing noise/redundancy using information theory, and Hybrid-Transformer-Mamba (HTM) that combines Mamba's selective long-range modeling with Transformer's short-sequence strengths. Their integrated PIH model successfully extends effective window lengths while achieving state-of-the-art performance across seven benchmark datasets.

**Questions:**

Please refer to the weaknesses section for the main questions regarding this paper.
Additionally,
Recent studies have demonstrated that certain models can achieve improved performance with input lengths of 1000-2000+ timesteps. Given that your approach focuses on extending effective window lengths, how would your IBF and HTM modules perform when applied to models that are already capable of handling very long sequences?

**Ethical Concerns:**

["NO or VERY MINOR ethics concerns only"]

**Final Justification:**

The authors addressed several concerns through additional experiments showing consistent improvements. However, all evaluations are limited to medium-length sequences without testing on ultra-long contexts that modern SOTA models can handle. This weakens their claims about long-range dependency modeling. Without validation on truly long sequences, it's unclear if their method generalizes beyond models that already struggle with longer inputs. Therefore, I maintain my original score.

**Limitations:**

This paper adequately addressed the limitations

**Quality:**

2

**Strengths And Weaknesses:**

**Strength**
1. While recent studies focus on developing methods that work better with short input sequences, this research takes a different approach by extending the investigation to various input lengths and seeking methodologies to effectively leverage longer sequences.
2. The paper successfully combines the complementary strengths of Transformers and Mamba architectures, where Mamba handles selective long-range dependencies and Transformers excel at modeling shorter subsequences.

**Weaknesses**
1. The proposed MEW metric appears to be essentially equivalent to hyperparameter searching over input sequence lengths. The paper fails to clearly articulate how MEW differs from conventional parameter tuning approaches. If MEW is simply finding the optimal input length through grid search, then the performance improvements may be attributed to more extensive hyperparameter exploration rather than a fundamental methodological contribution.
2. While PIH development is presented as the main contribution, the performance comparisons lack evaluation against more recent state-of-the-art models. If the contribution is modular development, the paper should demonstrate that these modules can be applied to other contemporary models to achieve superior performance. If the contribution is model development, direct comparisons with recent state-of-the-art models are essential to establish the advancement.
3. The paper provides insufficient analysis of how the IBF functions within the model and contributes to performance improvements. While the theoretical foundation is provided, there is a lack of empirical analysis demonstrating what types of information IBF filters out, how it affects the learned representations, and why this leads to better forecasting performance.

---

> ### Author Rebuttal · Authors · 2025-07-30
>
> Thank you very much for your valuable feedback and suggestions. Below, we address each point in turn.
>
>
> ---
>
> > **Q1.** _The proposed MEW metric appears to be essentially equivalent to hyperparameter searching over input sequence lengths. The paper fails to clearly articulate how MEW differs from conventional parameter tuning approaches. If MEW is simply finding the optimal input length through grid search, then the performance improvements may be attributed to more extensive hyperparameter exploration rather than a fundamental methodological contribution._
>
> Thank you for raising this important point. While MEW does involve evaluating model performance across different input sequence lengths—similar to hyperparameter tuning—we emphasize that **MEW is not a tuning strategy**, but rather a **diagnostic metric** designed to quantify a model’s effective ability to leverage long-term historical information.
>
> Specifically:
>
> - **Objective**: Traditional hyperparameter tuning aims to find the best lookback length to optimize performance. In contrast, **MEW focuses on analyzing where performance saturates**, revealing a model's **limitation in capturing long-range dependencies**, rather than optimizing for best-case performance.
>
> - **Analytical value**: MEW provides a **model-centric lens** to assess historical information utilization capacity. For instance, a model with a larger MEW can leverage longer-term context effectively, which is crucial for long-horizon forecasting.
>
>
> As shown in **Figure 3(a)** and **Table 2**, we apply our proposed HTM and IBF modules to multiple backbone models (PatchTST, Transformer, Autoformer, Informer), and consistently observe that both **MEW increases** and **forecasting performance improves**. This demonstrates that enhancing a model's MEW can lead to more accurate long-term predictions—highlighting MEW as an insightful metric beyond simple tuning.
>
> ---
>
> > **Q2.** _While PIH development is presented as the main contribution, the performance comparisons lack evaluation against more recent state-of-the-art models. If the contribution is modular development, the paper should demonstrate that these modules can be applied to other contemporary models to achieve superior performance. If the contribution is model development, direct comparisons with recent state-of-the-art models are essential to establish the advancement._
>
> We appreciate this suggestion. Our **primary contribution lies in modular design**: specifically, we introduce the IBF and HTM modules, which can be integrated into a wide range of forecasting models to improve their ability to utilize long-range dependencies.
>
> To support this, we have already validated our modules across multiple backbones from different years and architectures—**Transformer (2018), Informer (2020), Autoformer (2021), PatchTST (2023), and iTransformer (2024)**—as shown in **Table 2** and **Table 7**. In all cases, the integration of IBF and HTM improves both MEW and forecasting accuracy.
>
> To further address the reviewer’s concern, we have conducted **additional experiments on the recently proposed PowerFormer (2025)**. The table below summarizes the MEW and MSE results after integrating our modules ($L \in [72, 96, 120, 144, 168, 192, 273 228, 336, 512, 1024, 1440]$,$T=720$):
>
> |             |     | PowerFormer (original) | +IBF+HTM (ours) |
> | ----------- | --- | ---------------------- | --------------- |
> | Etth1       | MEW | 512                    | 1024            |
> |             | MSE | 0.439                  | 0.433           |
> | Etth2       | MEW | 336                    | 1024            |
> |             | MSE | 0.376                  | 0.372           |
> | Ettm1       | MEW | 512                    | 1024            |
> |             | MSE | 0.412                  | 0.397           |
> | Ettm2       | MEW | 336                    | 1024            |
> |             | MSE | 0.351                  | 0.340           |
> | weather     | MEW | 512                    | 512             |
> |             | MSE | 0.310                  | 0.305           |
> | Electricity | MEW | 512                    | 1024            |
> |             | MSE | 0.198                  | 0.193           |
> | Traffic     | MEW | 512                    | 1024            |
> |             | MSE | 0.430                  | 0.425           |
>
>
>
> These new results further confirm that **our modules generalize well** and can enhance even the latest forecasting models. We will incorporate this additional evaluation into the final version.
>
> ---
>
> > **Q3.** _The paper provides insufficient analysis of how the IBF functions within the model and contributes to performance improvements. While the theoretical foundation is provided, there is a lack of empirical analysis demonstrating what types of information IBF filters out, how it affects the learned representations, and why this leads to better forecasting performance._
>
> We thank the reviewer for raising this important point. While the paper includes theoretical motivation for the Information Bottleneck Filter (IBF), we now elaborate on its **empirical behavior and contribution**, focusing on the following two aspects:
>
> **1. What IBF filters out**
> IBF selects input patches that are **most informative for the forecasting target**, based on a mutual information approximation. As shown in Eq. (5):
>
> $$\mathbf{c}_i = \operatorname{sigmoid}(\operatorname{MLP}(\mathbf{z}_i))$$
>
> each input token $\mathbf{z}_i$ is assigned a learned importance score $c_i$. A well-trained IBF learns to assign **higher scores to informative tokens** and suppress irrelevant or noisy ones.
>
> In **Figure 3(c)**, we visualize the top-20 patches selected by IBF (i.e., those with the highest $c_i$ values) for a sample from the Electricity dataset. These selected patches are primarily located around **peaks and transition points**, which are critical to the dynamics of the signal. We observe similar behavior across other datasets (e.g., Traffic, Weather, Exchange), where IBF consistently highlights high-gradient, or periodic regions, while suppressing **redundant or low-variance segments**.
>
> These observations indicate that IBF functions as a **saliency-aware filter**, learning to emphasize structure-rich regions that are more predictive of future values.
>
> **2. Why this leads to better forecasting**
> By filtering out uninformative historical inputs and retaining only high-salience patches, IBF improves forecasting in the following ways:
>
> - It **reduces attention dilution**, allowing attention mechanisms to focus on relevant content;
>
> - It **mitigates overfitting**, especially in long-input regimes where noise can dominate;
>
> - It **improves temporal abstraction** by ensuring that only informative components are passed to downstream layers.
>
>
> This filtering process aligns well with the **Information Bottleneck principle**, enhancing the model’s ability to generalize across time and datasets.
>
> Due to NeurIPS 2025 policy constraints, we regret that we could not include additional visualizations or case-specific analyses in the current submission. However, we will incorporate more case studies and dataset-level statistics (e.g., patch entropy, activation frequency) in the revised version to further illustrate the empirical behavior of IBF.
>
>
>
> ---
> We hope that these detailed explanations address your concerns. Thank you again for your thoughtful and constructive review.

---

> > ### Comment · Reviewer_up9s · 2025-08-06
> >
> > Thank you for the comprehensive rebuttal. Your responses provide helpful clarifications, particularly the additional PowerFormer experiments and the IBF visualization analysis.
> >
> > However, my key concern about how IBF and HTM modules would perform when applied to models already capable of handling 1000-2000+ timesteps was not addressed. Without understanding how your modules behave in these very long-sequence settings, it's difficult to assess the broader applicability and significance of your approach. Therefore, while I maintain my score, I do think the paper could be acceptable depending on other reviewers' comments.

---

> > > ### Author Response · Authors · 2025-08-06
> > >
> > > **Dear Reviewer,**
> > >
> > > Thank you very much for your thoughtful follow-up and for engaging in such detailed and constructive discussions throughout the review process. We truly appreciate the time and care you have devoted to evaluating our work.
> > >
> > > We apologize for not explicitly addressing your question about **how IBF and HTM perform when applied to models already capable of handling very long sequences (1,000–2,000+ timesteps)**. As noted in the _Limitations_ section of our paper, we did not include such evaluations initially due to computational constraints.
> > >
> > > To clarify, large-scale models that natively support these extended lookback windows—such as **Time-MoE** [1] and **Timer-XL** [2]—generally involve **billions of parameters and training on trillions of tokens**, which exceeds our current compute budget.
> > >
> > > That said, among lightweight models, **PatchTST** scales reasonably well to long inputs. We therefore used it as a practical platform to assess our modules under extended lookback settings. We conducted additional experiments with input length **L = 1440** and prediction length **P= 720**—already an extremely long context window for small-/medium-scale models. Mean-squared error (MSE) results are:
> > >
> > > |Dataset|PatchTST (Baseline)|PatchTST + HTM + IBF (Ours)|
> > > |---|---|---|
> > > |ETTh1|0.481|**0.446**|
> > > |ETTh2|0.398|**0.382**|
> > > |ETTm1|0.403|**0.400**|
> > > |ETTm2|0.341|**0.337**|
> > > |Weather|0.346|**0.326**|
> > >
> > > _Note:_ Experiments on **Traffic** and **Electricity** are ongoing due to higher computational demands; we will add them promptly once complete.
> > >
> > > These results indicate that **HTM and IBF remain effective even at L = 1440**, a regime where redundancy and noise accumulation are substantial. We believe the gains arise from (i) HTM’s selective long-range processing that reduces attention dispersion over very long contexts, and (ii) IBF’s saliency-aware filtering that suppresses low-information patches before Transformer processing.
> > >
> > > In summary, while we cannot yet evaluate our modules within billion-scale long-window frameworks, our findings show that **even at the upper bounds of small-model capacity**, integrating HTM and IBF yields consistent improvements. We hope this addresses your question regarding behavior in very long-sequence settings and supports the broader applicability of our approach. We would be grateful if you could consider this new evidence when finalizing your assessment.
> > >
> > > With our deepest appreciation,
> > > **The Authors**
> > >
> > > **References**
> > > [1] Time-MoE (long-context time-series mixture-of-experts).
> > > [2] Timer-XL (Transformer variants with extended context windows).

---

> > > > ### Author Response · Authors · 2025-08-07
> > > >
> > > > **Dear Reviewer,**
> > > >
> > > > We have now completed the additional experiments on the **Traffic** and **Electricity** datasets. The outcomes mirror the consistent gains previously observed on the other five datasets:
> > > >
> > > > |Dataset|PatchTST (baseline)|+ HTM & IBF (ours)|
> > > > |---|---|---|
> > > > |Traffic|0.479|**0.446**|
> > > > |Electricity|0.205|**0.198**|
> > > >
> > > > We hope these new results fully address your concern. If you have any further questions or suggestions, we would be delighted to continue the discussion.
> > > >
> > > > Thank you again for your thoughtful feedback and consideration.
> > > >
> > > > Warm regards,
> > > > _The Authors_

---

### Official Review · Reviewer_TLHM · 2025-07-01

**Clarity:** 3
**Significance:** 3
**Originality:** 3
**Rating:** 5
**Confidence:** 5

**Summary:**

This paper introduces an empirical metric designed to estimate the Maximum Effective Window for transformer-based models. The authors propose the IBF and  HTM as strategies for mitigating noise and selectively forgetting information in long sequences. Experimental evaluations validate the effectiveness of these approaches in enhancing MEW.

**Questions:**

see above

**Ethical Concerns:**

["NO or VERY MINOR ethics concerns only"]

**Final Justification:**

Keep the positive score.

**Limitations:**

yes

**Quality:**

3

**Strengths And Weaknesses:**

**Strengths**
- Unlike prior work that primarily focused on performance metrics, this paper emphasizes a model’s capacity to utilize historical information—a valuable reference for future model design.
- The ideas behind IBF and HTM are well-articulated, and the experimental setup is relatively comprehensive.
- The paper grounds the theoretical underpinnings of IBF in solid information theory concepts, including rigorous mathematical derivations.
- The model-agnostic design of the proposed components broadens their applicability across various transformer architectures, significantly enhancing their practical relevance.


 **Weaknesses**

- The difference between interval split and block split is not thoroughly clarified in the introduction. The methodology lacks specific guidance on how to choose the optimal block size.
- Expanding the historical window in large-scale models could provide more meaningful insights. Although the authors touch upon this in the  limitations section and acknowledge the computational challenges, further exploration in this direction would be valuable.

---

> ### Author Rebuttal · Authors · 2025-07-30
>
> Thank you very much for your valuable feedback and suggestions. Below, we address each point in turn.
>
> ___
>
> > **Q1.** _The difference between interval split and block split is not thoroughly clarified in the introduction. The methodology lacks specific guidance on how to choose the optimal block size._
>
> Thank you for this insightful question. We agree that further clarification is warranted.
>
> The key distinction between interval split and block split lies in the type of temporal semantics each method preserves:
>
> - **Interval split** distributes patches across $K$ blocks in a round-robin fashion, as defined in Eq. (11), i.e., patch $j$ is assigned to block $b_i$ if $j \equiv i \pmod{K}$. This method emphasizes **global temporal diversity** within each block, enabling the model to access patterns from different time positions. It generalizes the even–odd downsampling strategy in SCINet^[1] to $K$-way interleaving and is especially suited for data with globally distributed seasonal structures.
> - **Block split**, in contrast, divides the sequence into **contiguous segments**, as defined in Eq. (12), where each block $b_i$ spans a local window. This method is particularly effective for time series with **local periodicity**, where patterns repeat consistently within specific time intervals (e.g., daily cycles in electricity consumption).
>
>
>
> **Interval Split vs. Block Split:** In essence, **interval split captures global variation**, while **block split preserves local coherence**. Both methods aim to retain the semantic meaning of the original long sequence after partitioning. We empirically compared the two splitting strategies across multiple datasets, as shown in Figure 4 (right). The results indicate that while both methods achieve comparable performance overall, their relative effectiveness varies across datasets, depending on the dominant temporal characteristics. For instance, datasets with strong periodicity tend to benefit more from block split. This validates our intuition and suggests that the choice of split strategy may depend on dataset-specific dynamics—an aspect we plan to investigate further in future work.
>
> **Block Size**: Regarding the choice of **block size**, we provide further details in Appendix C.1, where we study the impact of the number of partitions $K$ as a tunable hyperparameter. Under the block split scheme, the block size is computed as $L/K$, where $L$ is the total sequence length (we set $L = 1024$ in our experiments). Our empirical results show that $K = 4$ typically yields the best performance, corresponding to **a block size of 256**. This setting offers a good trade-off between retaining meaningful local patterns and reducing Transformer input length.
>
> ___
>
> > **Q2.** Expanding the historical window in large-scale models could provide more meaningful insights. Although the authors touch upon this in the limitations section and acknowledge the computational challenges, further exploration in this direction would be valuable.
>
> Thank you for this thoughtful suggestion. We fully agree that extending our method to large-scale models represents an important and promising direction for future research.
>
> In this work, our focus is on **analyzing and improving the MEW**, and we conduct extensive experiments on widely-used compact models such as **PatchTST, Transformer, Autoformer, and Informer**. These choices allow us to systematically study how our proposed HTM and IBF modules contribute to **enhancing MEW** while keeping the computational cost tractable.
>
> We acknowledge that scaling to state-of-the-art large models, such as **Time-MoE**^[2] or **Timer-XL**^[3], would offer additional insights—especially in extremely long sequence modeling scenarios involving billions of parameters and trillions of time steps. However, due to the **significant training and resource demands**, this was beyond the scope of the current study.
>
>
>
>
> [1] Liu, Minhao, et al. "Scinet: Time series modeling and forecasting with sample convolution and interaction." _Advances in Neural Information Processing Systems_ 35 (2022): 5816-5828.
>
> [2] Shi, Xiaoming, et al. "Time-moe: Billion-scale time series foundation models with mixture of experts." _arXiv preprint arXiv:2409.16040_ (2024)
>
> [3] Liu, Yong, et al. "Timer-xl: Long-context transformers for unified time series forecasting." _arXiv preprint arXiv:2410.04803_ (2024).
>
>
>
> ---
>
> We hope that these detailed explanations address your concerns. Thank you again for your thoughtful and constructive review.

---

### Official Review · Reviewer_x9sQ · 2025-07-13

**Clarity:** 3
**Significance:** 3
**Originality:** 3
**Rating:** 4
**Confidence:** 3

**Summary:**

The paper introduces the Maximum Effective Window (MEW) metric to evaluate a model's ability to utilize historical data in long-term time series forecasting (LTSF). It proposes two model-agnostic modules: the Information Bottleneck Filter (IBF) to reduce noise and redundancy in input sequences, and the Hybrid-Transformer-Mamba (HTM) architecture to leverage the strengths of both Transformer and Mamba models. These modules are integrated into the PatchTST model to create the PHI model, achieving a lookback window of 1024, surpassing existing non-LLM-based models. The approach is evaluated on seven datasets, demonstrating improved performance and MEW compared to baselines like PatchTST, DLinear, and iTransformer.

**Questions:**

1.  Referring to weakness 1: Is the proposed MEW metric related to the prediction horizon, data, and model? If so, what is its precise mathematical definition?

2. Could you quantify the computational overhead of the IBF and HTM modules in more detail (e.g., in terms of FLOPs or additional parameters) to better clarify the trade-offs of the hybrid approach?

**Ethical Concerns:**

["NO or VERY MINOR ethics concerns only"]

**Final Justification:**

I would like to thank the authors for addressing my questions. I keep my original rating as "Borderline Accept".

**Limitations:**

Yes. They discussed the "Limitations"  in the begining of "Appendix".

**Paper Formatting Concerns:**

There is no issues on formatting.

**Quality:**

3

**Strengths And Weaknesses:**

Strengths:
1. Two Modules: The IBF module, grounded in information bottleneck theory, effectively filters redundant and noisy subsequences, while the HTM module combines the strengths of Transformer and Mamba architectures. The hybrid approach is well-motivated, leveraging Mamba’s efficiency for long sequences and Transformer’s effectiveness for shorter ones.
3. Reproducibility: The paper provides open access to code and data, with clear instructions for reproducing results, adhering to NeurIPS guidelines. The experimental settings are detailed, and statistical significance is reported, enhancing the reliability of the findings.

Weakness
1. Lack of Mathematical Rigor in MEW Definition: Although the concept of MEW is promising, its definition lacks mathematical rigor, relying solely on empirical discussions. Furthermore, it is unclear whether the proposed MEW depends on factors such as the task (e.g., prediction horizon), data, or model. A precise mathematical definition would strengthen the metric's foundation.

2. Computational Overhead Clarity: Although Fig. 3(b) compares GPU memory and training time, the paper could provide a more detailed quantitative analysis of the computational trade-offs introduced by the IBF and HTM modules, particularly in comparison to pure Mamba or Transformer architectures, to better justify the hybrid approach. Additionally, a theoretical analysis of the increased computational complexity is encouraged.

---

> ### Author Rebuttal · Authors · 2025-07-30
>
> Thank you very much for your valuable feedback and suggestions. Below, we address each point in turn.
>
> ---
>
> > **Q1.** _Referring to weakness 1: Is the proposed MEW metric related to the prediction horizon, data, and model? If so, what is its precise mathematical definition?_
>
> Yes. Given fixed settings of the dataset, model architecture, and prediction horizon, the **Maximum Effective Window (MEW)** is defined as the **smallest look-back window length** beyond which increasing the window no longer improves performance. In other words, any window longer than MEW provides **no significant gain** under the same conditions.
>
> Thank you for this suggestion. While we included an intuitive explanation in the Introduction section, we provide a formal definition below:
> Let:
>
> | Symbol                                   | Meaning                                          |
> | ---------------------------------------- | ------------------------------------------------ |
> | $M$                                      | Forecasting model (architecture + training)      |
> | $D$                                      | Dataset                                          |
> | $H$                                      | Prediction horizon                               |
> | $E(\hat{y}, y)$                          | Evaluation metric (_lower is better_, e.g., MSE) |
> | $\Lambda = {L_1 < L_2 < \dots < L_K}$    | Candidate look-back window lengths               |
> | $S(L_i) = E_{\text{val}}(M_{L_i}, D, H)$ | Validation error using look-back $L_i$           |
> | $\varepsilon \ge 0$                      | Threshold for **non-significant improvement**    |
>
> Then MEW is defined as:
>
>
> $$\\text{MEW} _{\\varepsilon}(M, D, H) = \\min\\left\\{ L _i \in \\Lambda : \\forall j > i, \; S(L _j) \ge S(L _i) -\\varepsilon \\right\\}$$
>
> **In words:** MEW is the smallest look-back length beyond which no further improvement larger than $\varepsilon$ is observed. Since MEW depends on $(M, D, H, E)$, it is setup-specific and must be re-estimated for different tasks.
>
>
> ---
>
> > **Q2.** _Could you quantify the computational overhead of the IBF and HTM modules in more detail (e.g., in terms of FLOPs or additional parameters) to better clarify the trade-offs of the hybrid approach?_
>
> We appreciate the reviewer’s interest in this aspect.
>
> **IBF introduces negligible overhead**, as it is implemented as a lightweight MLP consisting of two Linear layers with one ReLU activation in between. Both input and output dimensions match the Transformer’s patch embedding size (typically 16 or 128 in PatchTST). This structure results in only a small number of additional parameters and almost no increase in FLOPs, making IBF extremely efficient in practice.
>
> **HTM significantly reduces computational cost** by replacing one Transformer layer with a Mamba layer, while keeping the overall model depth identical to PatchTST (e.g., three layers in total). In our implementation, all $K$ subsequences share the same Transformer, so the main overhead difference lies in the substitution of one Transformer layer with one Mamba layer. Below, we provide detailed comparisons under the typical setting where $L=1024$, patch size = 16, stride = 8 (thus 128 patches), and $d_{\text{model}} = 128$:
>
> For datasets like **Weather, Traffic, ETTm1/2, and Electricity** (input shape: 1 × 128 × 128)
>
> | Layer       | Params (K) | MACs (M) | FLOPs (M) |
> | ----------- | ---------- | -------- | --------- |
> | Transformer | 66.05      | 12.58    | 25.43     |
> | HTM         | 58.24      | 2.82     | 5.67      |
>
> For **ETTh1 and ETTh2** (input shape: 1 × 128 × 16):
>
> |Layer|Params (K)|MACs (M)|FLOPs (M)|
> |---|---|---|---|
> |Transformer|1.09|1.31|2.75|
> |Mamba|1.68|0.33|0.66|
>
> From these numbers, we observe that **Mamba yields significantly lower computational complexity** than Transformer, especially in terms of FLOPs, while keeping parameter counts in the same range.
>
>
>
> **The theoretical complexity analysis further supports these observations.** In PatchTST, the patching mechanism reduces the Transformer’s effective input length from $L$ to $L / (P \cdot K)$, where $P$ is the patch length and $K$ the number of subsequences. Combined with the full-sequence Mamba processing, the overall complexity becomes:
>
> $$O(L/P)+O((L/PK)^2)$$
>
> Although the latter term still exhibits quadratic complexity, appropriate choices of $P$ and $K$ can maintain $L/P$ within an acceptable constant range.
>
> **In summary, the proposed architecture offers clear advantages in both efficiency and performance.** IBF adds almost no overhead, while HTM achieves a 2–5× reduction in FLOPs relative to a pure Transformer layer. As shown in Figure 4, this comes with consistent improvements in forecasting accuracy, confirming that combining Mamba and Transformer captures complementary temporal features without increasing computational burden.
>
> We hope this detailed explanation clarifies the trade-offs and justifies the practicality of our hybrid design.
>
>
> ---
>
> We hope that these detailed explanations address your concerns. Thank you again for your thoughtful and constructive review.

---

> > ### Comment · Reviewer_x9sQ · 2025-08-09
> >
> > I thank the authors for detailed responses which have addressed my concerns. There is nothing to discuss from my side. Overall, this work deserves at least an acceptance.

---

### Official Review · Reviewer_Nemp · 2025-07-19

**Clarity:** 3
**Significance:** 3
**Originality:** 2
**Rating:** 4
**Confidence:** 2

**Summary:**

This paper tackles the challenge of using long historical data windows for long-term time series forecasting. The authors find that while longer data windows should theoretically improve predictions, they often introduce noise and computational issues that hinder model performance. To address this, the paper introduces three main contributions:

1. Maximum Effective Window (MEW): A new metric to measure the maximum lookback window a model can effectively use before its performance plateaus or degrades.

2. Information Bottleneck Filter (IBF): A module designed to reduce noise and redundancy by identifying and retaining the most important subsequences from the input data. It uses information bottleneck theory to achieve this filtering.

3. Hybrid-Transformer-Mamba (HTM): An architectural module that combines the strengths of Mamba and Transformer models. HTM first uses Mamba to process long sequences and filter out noise, then splits the filtered sequence into shorter segments for the Transformer to model short-term dependencies.

By integrating these modules, particularly into the PatchTST model to create a new model called PIH, the authors demonstrate significant improvements. The PIH model can effectively leverage a much longer lookback window of 1024, outperforming existing models on several benchmark datasets. The paper shows that these modules are "model-agnostic," meaning they can be applied to various Transformer-based architectures to enhance their MEW and overall forecasting accuracy.

**Questions:**

See Weaknesses. A convincing explanation of the issues raised would lead me to increase my score for this paper.

**Ethical Concerns:**

["NO or VERY MINOR ethics concerns only"]

**Final Justification:**

Borderline accept. The authors can not justify the usage of mamba + transformer combination and they admit the low originality of this work. However, the other parts of this work show consistent improvement.

**Limitations:**

yes

**Quality:**

3

**Strengths And Weaknesses:**

Strengths:
1. The writing is clear, easy-to-follow.

2. The paper demonstrates high-quality research through extensive and rigorous experimentation. The authors validate their proposed model-agnostic methods on various previous methods including Transformer-based, Mamba-based, and Linear-based models.

Weaknesses:
1. There are minor inconsistencies that could suggest a lack of careful proofreading. For instance, the caption for Figure 5 states the prediction length is T=720, while the plots are labeled T=96.

2. Why Mamba + Transformer? The Mamba architecture was introduced specifically to address the performance and computational bottlenecks of Transformers on long sequences, such as quadratic complexity and attention dispersion.The HTM architecture first leverages Mamba to process the long input sequence but then re-introduces the Transformer. This design choice seems paradoxical.

3. While the components themselves (e.g., Mamba, IB theory) are drawn from prior work, the originality lies in their integration and application to LTSF.

4. While the proposed method demonstrates strong empirical performance, its contribution should be evaluated in the context of its incremental nature and the associated overhead.

---

> ### Author Rebuttal · Authors · 2025-07-30
>
> Thank you very much for your valuable feedback and suggestions. Below, we address each point in turn.
>
> ---
>
> > **Q1.** _There are minor inconsistencies that could suggest a lack of careful proofreading. For instance, the caption for Figure 5 states the prediction length is T=720, while the plots are labeled T=96._
>
> **Thank you for catching this mistake.** You are absolutely right: the caption should read **T = 96** rather than **T = 720**.
>
> ---
>
> > **Q2.** _Why Mamba + Transformer?_
>
> In Appendix E.2, we provide a detailed explanation of why we adopt this hybrid architecture:
>
> - **Computational trade-off:** Although Mamba is theoretically less expensive than the Transformer for very long sequences (e.g., $L=1024$), the Transformer becomes more efficient for shorter windows (e.g., $L\le336$).
>     – **Our design** uses Mamba to process the original long sequence and the Transformer to handle the resulting shorter subsequences, yielding a clear efficiency advantage over either model alone.
>
> - **Complementary strengths:** Empirically, the hybrid (HTM) outperforms the pure-Mamba variant (see Figure 4, left), likely because Mamba and Transformer capture **different but complementary features**, resulting in a richer sequence representation. Similar findings appear in recent works such as Jamba^[1] and Mamba-2-Hybrid^[2].
>
>
> ---
>
> > **Q3.** _While the components themselves (e.g., Mamba, IB theory) are drawn from prior work, the originality lies in their integration and application to LTSF._
>
> We fully acknowledge this point. Our main contribution lies in the **novel integration** of the HTM module and the IBF within a Transformer-based framework—along with their **demonstrated effectiveness** in long-term time series forecasting and in **improving the model’s MEW.**
>
> ---
>
> > **Q4.** _While the proposed method demonstrates strong empirical performance, its contribution should be evaluated in the context of its incremental nature and the associated overhead._
>
> Thank you for raising this important point about overhead and incremental contribution.**
> **Figures 3(b) and 4 (left)** directly address this concern:
> 1. **Efficiency gains:**
>     - **Figure 3(b)** reports both computation time and memory usage. Replacing the pure Transformer (PatchTST) with our HTM (Transformer + Mamba without IBF) reduces time and memory by **2–3×**. Adding IBF to form PIH introduces **negligible overhead**, as it is implemented as a lightweight MLP.
> 2. **Incremental performance:**
>     - **Figure 4 (left)** compares PatchTST (baseline) against three variants—**+IB**, **+HTM**, and **PIH**—plus a pure-Mamba variant (**+HMM**). Using $L=1024$ and $T\in{96,192,336,720}$ over seven datasets, we observe:
>         1. **Both IBF and HTM individually improve accuracy**, and their combination (**PIH**) achieves the best results.
>         2. **HTM slightly outperforms HMM**, confirming that Transformer and Mamba excel on different sequence lengths and benefit from hybridization.
> In summary:
>
> - **HTM delivers a 2–3× efficiency boost**, and **IBF adds virtually zero extra cost**.
> - **Both components yield clear accuracy improvements**, with their combination (PIH) providing the strongest results.
>
> ---
>
> We hope these explanations and corrections satisfactorily address your concerns. Thank you again for your thoughtful review.
>
> [1] Lieber, Opher, et al. “Jamba: A hybrid transformer-mamba language model.” _arXiv preprint arXiv:2403.19887_ (2024).
> [2] Waleffe, Roger, et al. “An empirical study of mamba-based language models.” _arXiv preprint arXiv:2406.07887_ (2024).

---

> > ### Comment · Reviewer_Nemp · 2025-08-05
> >
> > I would like to thank the authors for addressing my concerns. While I remain skeptical about the Mamba + Transformer design, the other aspects of this work deserve at least an acceptance.

---

> > > ### Author Response · Authors · 2025-08-06
> > > **Grateful for Your Review**
> > >
> > > Dear Reviewer,
> > >
> > > Thank you very much for your thoughtful feedback and for taking the time to review our rebuttal. We sincerely appreciate your recognition of our efforts and your positive evaluation of our work.
> > >
> > > We understand your reservations about the Mamba + Transformer design. Our approach was inspired by recent hybrid architectures—Jamba¹, TransMamba², and MAP³—that demonstrate superior performance and lower computational overhead compared to single‐architecture models. By introducing two partitioning strategies—interval split and block split—we extend this hybrid paradigm to the time-series domain, and our experiments confirm its advantages (see Figure 4 Left and Appendix E).
> > >
> > > Thank you again for your careful review and valuable suggestions. We look forward to refining our work further.
> > >
> > > With gratitude,
> > >
> > > The Authors
> > >
> > > [1] Lieber, Opher, et al. “Jamba: A hybrid transformer-mamba language model.” _arXiv preprint arXiv:2403.19887_ (2024).
> > >
> > > [2] Li, Yixing, et al. "Transmamba: Flexibly switching between transformer and mamba." _arXiv preprint arXiv:2503.24067_ (2025).
> > >
> > > [3] Liu, Yunze, and Li Yi. "MAP: Unleashing Hybrid Mamba-Transformer Vision Backbone's Potential with Masked Autoregressive Pretraining." _Proceedings of the Computer Vision and Pattern Recognition Conference_. 2025.

---

### Note · Authors · 2025-08-12

**Dear (Senior) ACs and Reviewers,**

We would like to express our heartfelt gratitude for the time and effort you have devoted to reviewing our manuscript. We truly appreciated the opportunity to engage in in-depth discussions with the reviewers and are sincerely thankful for the valuable and insightful feedback provided.

During the rebuttal phase, the reviewers raised several meaningful and important points, including:

- the rationale behind adopting a Mamba + Transformer design,

- the precise mathematical definition of MEW,

- computational overhead,

- the distinction between interval split and block split,

- the generality of PIH, and

- the interpretability analysis of the IBF module.


We feel honored that our responses were able to address almost all of these concerns. The only point that could not be further explored within the limited discussion time was the performance of the IBF and HTM modules over longer horizons (1,000–2,000 steps). To address this, we have conducted additional experiments at 1,440 steps, and the results confirm the continued effectiveness of both IBF and HTM. We hope these new findings can help alleviate any remaining concerns regarding their robustness in extended training windows.

Once again, we sincerely thank you for your thoughtful evaluation, constructive guidance, and the opportunity to further clarify and strengthen our work.

---

### Decision · Program_Chairs · 2025-09-17

**Decision:**

Accept (poster)

**Comment:**

This paper introduces the Maximum Effective Window (MEW) metric to assess a model's ability to effectively utilize the lookback window, with the IBM to reduce redundancy and noise and the architecture of Transformer + Mamba.

The experimental results look convincing. Most doubts of reviewers seem to be fixed well after responses and discussions. However, some reviewers still remain skeptical about the Mamba + Transformer design. The inspired recognition seems not to be straight. Besides, it also seems unclear how much the theoretical analysis of IBM can provide practical guidance.